# MMD GAN with Random-Forest Kernels Conference Submissions

## Abstract

In this paper, we propose a novel kind of kernel, random forest kernel, to enhance the empirical performance of MMD GAN. Different from common forests with deterministic routings, a probabilistic routing variant is used in our innovated random-forest kernel, which is possible to merge with the CNN frameworks. Our proposed random-forest kernel has the following advantages: From the perspective of random forest, the output of GAN discriminator can be viewed as feature inputs to the forest, where each tree gets access to merely a fraction of the features, and thus the entire forest benefits from ensemble learning. In the aspect of kernel method, random-forest kernel is proved to be characteristic, and therefore suitable for the MMD structure. Besides, being an asymmetric kernel, our random-forest kernel is much more flexible, in terms of capturing the differences between distributions. Sharing the advantages of CNN, kernel method, and ensemble learning, our random-forest kernel based MMD GAN obtains desirable empirical performances on CIFAR-10, CelebA and LSUN bedroom data sets. Furthermore, for the sake of completeness, we also put forward comprehensive theoretical analysis to support our experimental results.

## 1 Introduction

Generative adversarial nets (GANs; Goodfellow et al., 2014) are well-known generative models, which largely attribute to the sophisticated design of a generator and a discriminator which are trained jointly in an adversarial fashion. Nowadays GANs are intensely used in a variety of practical tasks, such as image-to-image translation (Tang et al., 2019; Mo et al., 2019); 3D reconstruction (Gecer et al., 2019); video prediction (Kwon & Park, 2019); text-to-image generation (Zhu et al., 2019); just to name a few.

However, it's well-known that the training of GANs is a little tricky, see e.g. (Salimans et al., 2016). One reason of instability of GAN training lies in the distance used in discriminator to measure the divergence between the generated distribution and the target distribution. For instance, concerning with the Jensen-Shannon divergence based GANs proposed in Goodfellow et al. (2014), Arjovsky & Bottou (2017) points out that if the generated distribution and the target distribution are supported on manifolds where the measure of intersection is zero, Jensen-Shannon divergence will be constant and the KL divergences be infinite. Consequently, the generator fails to obtain enough useful gradient to update, which undermines GAN training. Moreover, two non-overlapping distributions may be judged to be quite different by the Jensen-Shannon divergence, even if they are nearby with high probability.

As a result, to better measure the difference between two distributions, Integral Probability Metrics (IPM) based GANs have been proposed. For instance, Arjovsky et al. (2017) utilizes Wasserstein distance in GAN discriminator, while Li et al. (2017) adopts maximum mean discrepancy (MMD), managing to project and discriminate data in reproducing kernel Hilbert space (RKHS). To mention, the RKHS with characteristic kernels including Gaussian RBF kernel (Li et al., 2017) and rational quadratic kernel (Bińkowski et al., 2018) has strong power in the discrimination of two distributions, see e.g. (Sriperumbudur et al., 2010).

In this paper, inspired by non-linear discriminating power of decision forests, we propose a new type of kernel named random-forest kernel to improve the performance of MMD GAN discriminator. In order to fit with back-propagation training procedure, we borrow the decision forest model with

stochastic and differentiable decision trees from Kontschieder et al. (2015) in our random-forest kernel. To be specific, each dimension of the GAN discriminator outputs is randomly connected to one internal node of a soft decision forest, serving as the candidate to-be-split dimension. Then, the tree is split with a soft decision function through a probabilistic routing. Other than the typical decision forest used in classification tasks where the value of each leaf node is a label, the leaf value of our random forest is the probability of a sample $x_i$ falling into a certain leaf node of the forest. If the output of the discriminator is denoted as $h_{\theta_N}(x_i)$ and the probability output of the $t$-th tree is denoted as $\mu^t(h_{\theta_N}(x_i); \theta_F)$, the random forest kernel $k_{RF}$ can be formulated as

$$k_{RF}(x_i, x_j; \theta_F) = \frac{1}{T} \sum_{t=1}^{T} \left\langle \mu^t(h_{\theta_N}(x_i); \theta_F), \mu^t(h_{\theta_N}(x_j); \theta_F) \right\rangle,$$

where $T$ is the total number of trees in the forest, $\theta_N$ and $\theta_F$ denote the parameters of the GAN discriminator and the random forest respectively.

Recall that random forest and deep neural networks are first combined in Kontschieder et al. (2015), where differentiable decision tree model and deep convolutional networks are trained together in an end-to-end manner to solve classification tasks. Then, Shen et al. (2017) extends the idea to label distribution learning, and Shen et al. (2018) makes further extensions in regression regime. Moreover, Zuo & Drummond (2017) , Zuo et al. (2018) and Avraham et al. (2019) also introduce deep decision forests. Apart from the typical ensemble method that averages the results across trees, they aggregate the results by multiplication. As for the combination of random forest and GAN, Zuo et al. (2018) introduce forests structure in GAN discriminator, combining CNN network and forest as a composited classifier, while Avraham et al. (2019) uses forest structure as one of non-linear mapping functions in regularization part.

On the other hand, in the aspect of relationship between random forest and kernel method, Breiman (2000) initiates the literature concerning the link. He shows the fact that a purely random tree partition is equivalent to a kernel acting on the true margin, of which form can be viewed as the probability of two samples falling into the same terminal node. Shen & Vogelstein (2018) proves that random forest kernel is characteristic. Some more theoretical analysis can be found in Davies & Ghahramani (2014), Arlot & Genuer (2014), Scornet (2016). However, despite their theoretical breakthroughs, forest decision functions used in these forest kernels are non-differentiable hard margins rather than differentiable soft ones, and thus cannot be directly used in back propagation regime.

To the best of our knowledge, MMD GAN with our proposed random-forest kernel is the first to combine random forest with deep neural network in the form of kernel MMD GAN. Through theoretical analysis and numerical experiments, we evaluate the effectiveness of MMD GAN with our random-forest kernel. From the theoretical point of view, our random-forest kernel enjoys the property of being characteristic, and the gradient estimators used in the training process of random-forest kernel GAN are unbiased. In numerical experiments, we evaluate our random-forest kernel under the setting of both the original MMD GAN (Li et al., 2017) and the one with repulsive loss (Wang et al., 2019). Besides, we also compare our random-forest kernel with Gaussian RBF kernel (Li et al., 2017), rational quadratic kernel (Bińkowski et al., 2018), and bounded RBF kernel (Wang et al., 2019). As a result, MMD GAN with our random-forest kernel outperforms its counterparts with respect to both accuracy and training stability.

This paper is organized as follows. First of all, we introduce some preliminaries of MMD GAN in Section 2. Then we review the concept of deep random forest and show how it is embedded within a CNN in 3.1. After that, random-forest kernels and MMD GAN with random-forest kernels are proposed in 3.2 and 3.3 respectively. Besides, the training techniques of MMD GAN with random-forest kernel are demonstrated in Section 3.4 and the theoretical results are shown in Section 3.5. Eventually, Section 4 presents the experimental setups and results, including the comparison between our proposed random-forest kernel and other kernels. In addition, all detailed theoretical proofs are included in the Appendices.

## 2  MMD GAN

GANs are recently introduced as a novel way of training a generative model. To learn a distribution $P_X$, we build an adversarial model composed of two parts, the generator $G$ and the discriminator $D$.

The generative model captures the data distribution $P_X$, by building a mapping function $G : \mathcal{Z} \to \mathcal{X}$ from a prior noise distribution $P_Z$ to data space. While the discriminative model $D : \mathcal{X} \to \mathbb{R}$ is used to distinguish generated distribution $P_Y$ from real data distribution $P_X$.

Taking $X, X' \sim P_X$ and $Y, Y' \sim P_Y := P_G(Z)$ where $Y := G(Z)$ and $Y' := G(Z')$, the squared MMD is expressed as

$$\mathrm{MMD}^2[P_X, P_Y] = \mathbb{E}_{X,X'} k(X, X') - 2\mathbb{E}_{X,Y} k(X, Y) + \mathbb{E}_{Y,Y'} k(Y, Y'). \tag{1}$$

The loss of generator and discriminator in MMD GAN proposed in Li et al. (2017) is:

$$\min_G L_G = \mathrm{MMD}^2[P_X, P_Y], \qquad \min_D L_D = -\mathrm{MMD}^2[P_X, P_Y].$$

Wang et al. (2019) proposed MMD GAN with repulsive loss, where the objective functions for G and D are:

$$\min_G L_G = \mathrm{MMD}^2[P_X, P_Y], \qquad \min_D L_D = \mathbb{E}_{X,X'} k(X, X') - \mathbb{E}_{Y,Y'} k(Y, Y').$$

Given i.i.d. samples $D_X = \{x_1, \ldots, x_n\} \sim P_X$ and $D_Y = \{G(z_1), \ldots, G(z_m)\} \sim P_Y$, we can write an unbiased estimator of the squared MMD in terms of $k$ as

$$\widehat{\mathrm{MMD}}_u^2[D_X, D_Y] = \frac{1}{n(n-1)} \sum_{i=1}^n \sum_{j \neq i}^n k(x_i, x_j)$$
$$+ \frac{1}{m(m-1)} \sum_{i=1}^m \sum_{j \neq i}^m k(y_i, y_j) - \frac{2}{nm} \sum_{i=1}^n \sum_{j=1}^m k(x_i, y_j). \tag{2}$$

When $k$ is a characteristic kernel, we have $\mathrm{MMD}^2[P_X, P_Y] \geq 0$ with equality applies if and only if $P_X = P_Y$. The best-known characteristic kernels are gaussian RBF kernel and rational quadratic kernel (Bińkowski et al., 2018).

## 3 RANDOM-FOREST KERNEL

In this section, we review a stochastic and differentiable variant of random forest and how it is embedded within a deep convolutional neural network proposed in Kontschieder et al. (2015). Then we propose random-forest kernel and we apply it in MMD GAN. We illustrate the advantages of our random-forest kernel, show the training technique of MMD GAN with random-forest kernel, and study its theoretical properties.

### 3.1 DEEP RANDOM FOREST

Suppose that a random forest consists of $T \in \mathbb{N}$ random trees. For the $t$-th tree in the forest, $t \in \{1, \ldots, T\}$, we denote $\mathcal{N}_t := \{d_j^t\}_{j=1}^{|\mathcal{N}_t|}$ as the set of its internal nodes and if $T$ trees have the same structure, then we have $|\mathcal{N}_t| = |\mathcal{N}|$, see Figure 1. Furthermore, we denote $\mathcal{L}_t$ as the set of its leaf nodes and $\theta_F^t$ as the parameters of the $t$-th tree.

Here we introduce the *routing function* $\mu_\ell^t(x; \theta_F^t)$ which indicates the probability of the sample $x$ falling into the $\ell$-th leaf node of the $t$-th tree. In order to provide an explicit form for the routing function $\mu_\ell^t(x; \theta_F^t)$, e.g. the thick black line in Figure 1, we introduce the following binary relations that depend on the tree structure: $\ell \swarrow d_j^t$ is true, if $\ell$ belongs to the left subtree of node $d_j^t$, and $\ell \searrow d_j^t$ is true, if $\ell$ belongs to the right subtree of node $d_j^t$. Moreover, let $p_j^t(x; \theta_F^t)$ be the decision function of the $j$-th internal node in the $t$-th tree, that is the probability of the sample $x$ falling into the left child of node $d_j^t$ in the $t$-th tree. Then, $\mu_\ell^t$ can be expressed by these relations as

$$\mu_\ell^t(x; \theta_F^t) = \prod_{d_j^t \in \mathcal{R}_\ell^t} p_j^t(x; \theta_F^t)^{\mathbf{1}_{\ell \swarrow d_j^t}} \left(1 - p_j^t(x; \theta_F^t)\right)^{\mathbf{1}_{\ell \searrow d_j^t}},$$

where $\mathcal{R}_\ell^t$ denotes the unique path from node 1 to node $\ell$ of the $t$-th tree.

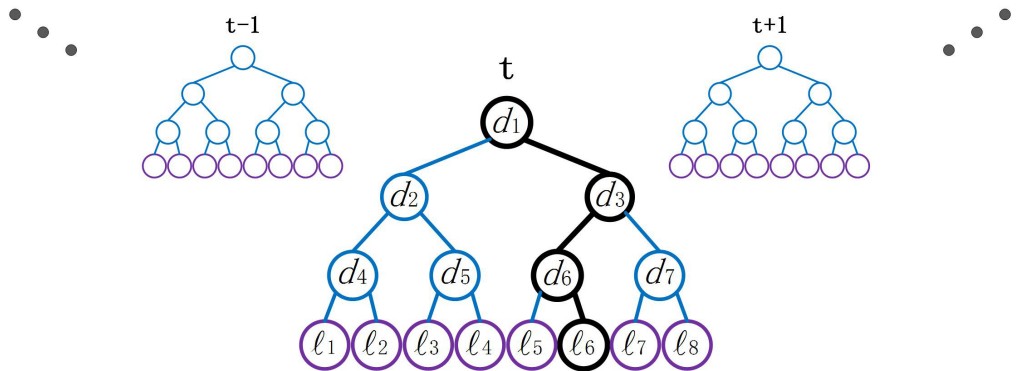

Figure 1: Example of a random forest with $T$ trees: blue nodes denote the internal nodes $\mathcal{N} := \{d_1, ..., d_7\}$ while purple nodes are the leaf nodes $\mathcal{L} := \{\ell_1, \cdots, \ell_8\}$. The black thick path illustrates the route of a sample $x$ falling into the $\ell_6$ leaf node of the $t$-th tree.

Now, let us derive the explicit form of the decision function $p_j^t(x; \theta_F^t)$. Here, to utilize the power of deep learning, we consider using the convolutional neural network to construct decision functions of random forest.

To be specific, given the parameter $\theta_N$ trained from a CNN network, we denote $h(\cdot; \theta_N)$ as the $d'$-dimension output of a convolutional neural network, which is the unit of the last fully-connected layer in the CNN, and $h_i(\cdot; \theta_N)$ is the $i$-th element of the CNN output. We denote $\mathcal{C} : \{1, \ldots, T|\mathcal{N}|\} \to \{1, \ldots, d'\}$ as the *connection function*, which represents the connection between the internal node $d_j^t$ and the former CNN output $h_i$. Note that during the whole training process, the form of the connection function $\mathcal{C}$ is not changed and every internal node $d_j^t$ is randomly assigned to an element $h_{\mathcal{C}(T(t-1)+j)}(\cdot; \theta)$. If we choose the sigmoid function $\sigma(x) = (1 + e^{-x})^{-1}$ as the decision function, and let the parameters of the $t$-th tree be $\theta_F^t := (w^t, b^t)$ with $w^t = (w_1^t, \ldots, w_{|\mathcal{N}|}^t)$ and $b^t = (b_1^t, \ldots, b_{|\mathcal{N}|}^t)$, then the decision functions $p_j^t$ delivering a stochastic routing can be defined as

$$p_j^t(x; \theta_N, \theta_F^t) = \sigma\left(w_j^t \cdot h_{\mathcal{C}(T(t-1)+j)}(x; \theta_N) + b_j^t\right).$$

For example, we have the probability $p_1^1(x; \theta_N, \theta_F^1) = \sigma(w_1^1 h_{\mathcal{C}(1)}(x; \theta_N) + b_1^1)$ for the node in the first layer, see Figure 2 for an explicit example of the random forest.

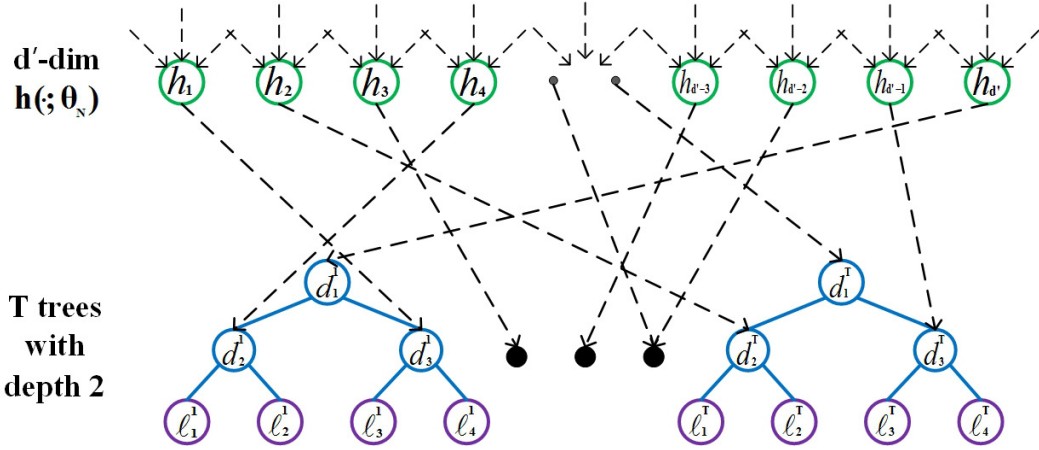

Figure 2: Example of a connection between CNN network $d'$-dimensional output $h(\cdot; \theta_N)$ and random forest with $T$ trees and depth 2. The internal node $d_3^1$ in the left tree has the probability $p_3^1(x; \theta_N, \theta_F^1) = \sigma(w_3^1 h_1(x; \theta_N) + b_3^1)$ and therefore the routing function of leaf node $\ell_4^1$ is $\mu_{\ell_4}^1(x; \theta_N, \theta_F^1) = \left(1 - \sigma(w_1^1 h_{d'}(x; \theta_N) + b_1^1)\right)\left(1 - \sigma(w_3^1 h_1(x; \theta_N) + b_3^1)\right).$

Every leaf node $\ell$ in each tree has a unique road $\mathcal{R}_\ell^t$ from node 1 to node $\ell$ with length $|\mathcal{R}_\ell^t| = \log_2(|\mathcal{L}_t|)$. Then, for the every leaf node $\ell$ of the $t$-th tree, we have

$$\mu_\ell^t(x) = \prod_{j \in \mathcal{R}_\ell^t} \sigma\big(w_j^t h_{C(T(t-1)+j)}(x; \theta_N) + b_j^t\big)^{\mathbf{1}_{j \in \mathcal{L}f}} \big(1 - \sigma(w_j^t h_{C(T(t-1)+j)}(x; \theta_N) + b_j^t)\big)^{1-\mathbf{1}_{j \in \mathcal{L}f}},$$

(3)

where $\mathcal{L}f$ denotes the set of all left son nodes of its father node.

## 3.2 RANDOM-FOREST KERNEL

Here, we propose the random-forest kernel as follows:

**Definition 1 (Random-Forest Kernel)** *Let $x, y$ be a pair of kernel input, let $\theta_F^t = (w^t, b^t)$ denotes the weights and bias of the $t$-th tree of the random forest, and $\theta_F := (\theta_F^t)_{t=1}^T$. The random-forest kernel can be defined as*

$$\begin{aligned}
k_{RF}(x, y; \theta_F) &= \frac{1}{T} \sum_{t=1}^T \sum_{\ell \in \mathcal{L}_t} \mu_\ell^t(x; \theta_F^t) \mu_\ell^t(y; \theta_F^t) \\
&= \frac{1}{T} \sum_{t=1}^T \big\langle \mu^t(x; \theta_F^t), \mu^t(y; \theta_F^t) \big\rangle \\
&= \frac{1}{T} \big\langle \mu^{(T)}(x; \theta_F), \mu^{(T)}(y; \theta_F) \big\rangle,
\end{aligned}$$

*where $\mathcal{L}_t$ denotes the set of leaf nodes in the $t$-th tree, $\mu^t = (\mu_\ell^t)_{\ell \in \mathcal{L}_t}$ and $\mu^{(T)} = (\mu^t)_{t=1}^T$.*

## 3.3 MMD GAN WITH RANDOM-FOREST KERNELS

We write

$$k(x, x') := k_{RF}\big(h_{\theta_N}(x), h_{\theta_N}(x'); \theta_F\big)$$

and introduce the objective functions of MMD GAN with random-forest kernel by

$$\begin{aligned}
\min_\psi L_G &= \mathrm{MMD}^2[P_X, P_Y] \\
&= \mathbb{E}_{X,X'} k(X, X') - 2\mathbb{E}_{X,Y} k(X, Y) + \mathbb{E}_{Y,Y'} k(Y, Y'), \\
\min_{\theta_N, \theta_F} L_D &= -\mathrm{MMD}^2[P_X, P_Y] + \mathcal{R} \\
&= -\mathbb{E}_{X,X'} k(X, X') + 2\mathbb{E}_{X,Y}[k(X, Y)] - \mathbb{E}_{Y,Y'} k(Y, Y') + \mathcal{R},
\end{aligned}$$

where $y = G_\psi(z)$, $z$ is noise vector, and $\mathcal{R}$ is the regularizer of random-forest kernel (the detail is shown in Section 3.4). In addition, the objective functions of MMD GAN with repulsive loss are

$$\begin{aligned}
\min_\psi L_G &= \mathrm{MMD}^2[P_X, P_Y] \\
&= \mathbb{E}_{X,X'} k(X, X') - 2\mathbb{E}_{X,Y} k(X, Y) + \mathbb{E}_{Y,Y'} k(Y, Y'), \\
\min_{\theta_N, \theta_F} L_D &= -\mathrm{MMD}^2[P_X, P_Y] + \mathcal{R} \\
&= \mathbb{E}_{X,X'} k(X, X') - \mathbb{E}_{Y,Y'} k(Y, Y') + \mathcal{R}.
\end{aligned}$$

Random-forest kernel MMD GAN enjoys the following advantages:

- Our proposed random-forest kernel used in MMD GAN benefits from *ensemble learning*. From the perspective of random forest, the output of MMD GAN discriminator $h(\cdot; \theta_N)$ can be viewed as feature inputs to the forest. To mention, each tree only gets access to merely a fraction of the features by random connection functions, and thus the entire forest benefits from ensemble learning.

- Our random-forest kernel MMD GAN enjoys the advantages of three powerful discriminative methods, which are CNN, kernel method, and ensemble learning. To be specific, CNN is good at extracting useful features from images; Kernel method utilize RKHS for discrimination; Ensemble learning utilizes the power of randomness and ensemble.
- Our proposed random-forest kernel has some good theoretical properties. In one aspect, random-forest kernel is proved to be characteristic in Shen & Vogelstein (2018). In another, in Section 3.5, the unbiasedness of the gradients of MMD GAN with random-forest kernel is proved.

## 3.4 RANDOM-FOREST KERNEL TRAINING TECHNIQUE

In Frosst & Hinton (2017), the authors mention that the tree may get stuck on plateaus if internal nodes always assign the most of probability to one of its subtree. The gradients will vanish because the gradients of the logistic-type decision function will be very closed to zero. In order to stabilize the training of random-forest kernel and avoid the stuck on bad solutions, we add penalty that encourage each internal node to split in a balanced style as Frosst & Hinton (2017) does, that is, we penalize the cross entropy between the desired 0.5, 0.5 average probabilities of falling into two subtrees and the actual average probability $\alpha$, $1 - \alpha$.

The actual average probability of the $i$-th internal node $\alpha_i$ is

$$\alpha_i = \frac{\sum_{x \in \Omega} P^i(x) p_i(x)}{\sum_{\mathbf{x} \in \Omega} P^i(x)},$$

where $P^i(x)$ is the routing probability of $x$ from root node to internal node $i$, $p_i(x)$ is the probability of $x$ falling into the left subtree of the $i$-th internal node, and $\Omega$ is the collection of mini-batch samples.

Then, the formulation of the regularizer is:

$$\mathcal{R}(\Omega) = -\lambda \sum_{i \in \text{Internal Nodes}} 0.5 \log(\alpha_i) + 0.5 \log(1 - \alpha_i),$$

where $\lambda$ is exponentially decayed with the depth of $d$ of the internal node by multiplying the coefficient $2^{-d}$, for the intuition that less balanced split in deeper internal node may increase the non-linear discrimination power.

When training random-forest kernel, a mini-batch of real samples $X$ and generated pictures $Y$ are both fed into the discriminator, and then $k(X, X)$, $k(X, Y)$ and $k(Y, Y)$ are calculated, where $k := k_{RF} \circ h(\cdot; \theta_N)$. Here, to notify, we find that the $\Omega$ in the regularizer formulation does matter in forest-kernel setting. It's better to calculate $\alpha_i$ and $\mathcal{R}(\Omega)$ in the case of $\Omega = X$, $\Omega = Y$, $\Omega = X \cup Y$ respectively, and then sum up three parts of regularizer as final regularizer $\mathcal{R}$.

Therefore, the formulation of regularizer $\mathcal{R}$ added in the training of random-forest kernel is

$$\mathcal{R} = \mathcal{R}(X) + 2 \cdot \mathcal{R}(X \cup Y) + \mathcal{R}(Y) \tag{4}$$

## 3.5 THEORETICAL RESULTS

In this subsection, we present our main theoretical results.

**Theorem 2 (Unbiasedness)** *Let $X$ be the true data on $\mathcal{X}$ with the distribution $\mathrm{P}_X$ and $Z$ be the noise on $\mathcal{Z}$ with the distribution $\mathrm{P}_Z$ satisfying $\mathbb{E}_{\mathrm{P}_X} \|X\|^\alpha < \infty$ and $\mathbb{E}_{\mathrm{P}_Z} \|Z\|^\alpha < \infty$ for some $\alpha \geq 1$. Moreover, let $G_\psi : \mathcal{Z} \to \mathcal{X}$ be a generator network, $h_{\theta_N} : \mathcal{X} \to \mathbb{R}^{d'}$ be a discriminator network, $k_{RF}$ be the random-forest kernel, and $\theta_D := (\theta_N, \theta_F)$ be the parameters of the GAN discriminator. Then, for $\mu$-almost all $\theta_D \in \mathbb{R}^{|\theta_D|}$ and $\psi \in \mathbb{R}^{|\psi|}$, there holds*

$$\mathbb{E}_{D_X \sim \mathrm{P}_X^m, D_Z \sim \mathrm{P}_Z^n} \left[ \partial_{\theta_D, \psi} \mathrm{MMD}_u^2(h_{\theta_N}(D_X), h_{\theta_N}(G_\psi(D_Z))) \right]$$
$$= \partial_{\theta_D, \psi} \mathrm{MMD}^2(h_{\theta_N}(\mathrm{P}_X), h_{\theta_N}(G_\psi(\mathrm{P}_Z))).$$

In other words, during the training process of Random-Forest Kernel MMD GAN, the estimated gradients of MMD with respect to the parameters $\psi$ and $\theta_D$ are unbiased, that is, the expectation and the differential are exchangeable.

## 4 EXPERIMENTS

In this section, we evaluate our proposed random-forest kernel in the setting of MMD GAN in (Li et al., 2017) and the MMD GAN with repulsive loss (Wang et al., 2019). To illustrate the efficacy of our random-forest kernel, we compare our random-forest kernel with Gaussian kernel (Li et al., 2017), rational quadratic kernel (Bińkowski et al., 2018) in the setting of the original MMD GAN loss, and compare our random-forest kernel with bounded Gaussian kernel (Wang et al., 2019) in the setting of MMD GAN with repulsive loss.

### 4.1 EXPERIMENTAL SETUP

#### 4.1.1 DATASETS

The experiments are evaluated on three benchmark datasets:

(1) the Cifar10 dataset of $32 \times 32$ pictures (Krizhevsky et al., 2009);

(2) the LSUN dataset of bedroom pictures resized to $64 \times 64$ (Yu et al., 2015);

(3) the CelebA dataset of celebrity faces pictures randomly cropped and resized to $160 \times 160$ (Liu et al., 2015). The images are scaled to range $[0, 1]$.

#### 4.1.2 KERNELS AND LOSSES

Under the setting of MMD GAN loss proposed in Li et al. (2017), we compare our proposed random-forest kernel with the Gaussian RBF kernel (Li et al., 2017) with mixture of kernel scales $\sigma$ and the Rational quadratic kernel (Bińkowski et al., 2018) with mixture of kernel scales $\alpha$. Moreover, under another setting of MMD GAN with repulsive loss proposed in Wang et al. (2019), we compare our random-forest kernel with the bounded RBF kernel (Wang et al., 2019).

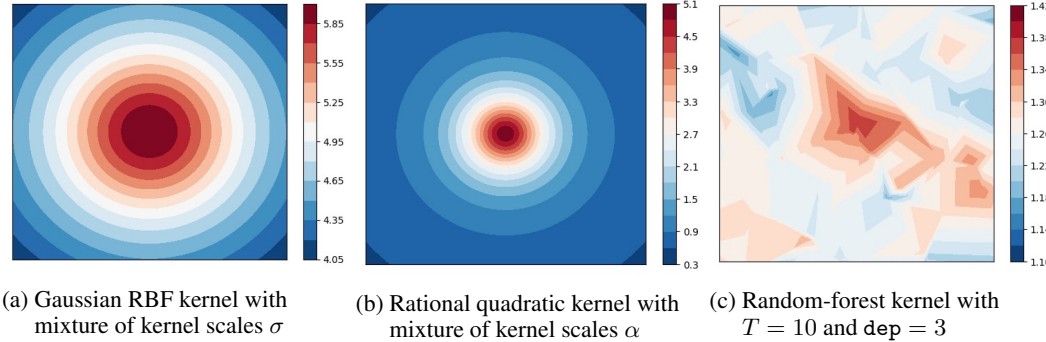

(a) Gaussian RBF kernel with mixture of kernel scales $\sigma$

(b) Rational quadratic kernel with mixture of kernel scales $\alpha$

(c) Random-forest kernel with $T = 10$ and dep $= 3$

Figure 3: Visualization of three kernels by drawing filled contours: (a) & (b) - A direct visualization of 2-dimensional kernels with reference to $(0, 0)$; (c) - A 2-dimensional visualization of the multi-dimensional random-forest kernel with the help of t-SNE (Maaten & Hinton, 2008). The details of visualization are shown in Appendix A.2.

As is shown in Figure 3, the shapes of the Gaussian RBF kernel and the rational quadratic kernel are both symmetric. However, the local structure of random-forest kernel (w.r.t reference points except 70-dimensional zero vector) is asymmetric and very complex. The asymmetry and complexity of random-forest kernel may be helpful to discriminate two distributions in MMD GAN training.

#### 4.1.3 NETWORK ARCHITECTURE

For dataset Cifar10 and dataset LSUN bedroom, DCGAN (Radford et al., 2016) architecture with hyper parameters from Miyato et al. (2018) is used for both generator and discriminator; and for dataset CelebA, we use a 5-layer DCGAN discriminator and a 10-layer ResNet generator. Further details of the network architecture are given in Appendix A.3. We mention that in all experiments, batch normalization (Ioffe & Szegedy, 2015) is used in the generator and spectral normalization (Miyato et al., 2018) is used in the discriminator. The hyper-parameter details of kernels used in

**Table 1:** Score evaluation results for three datasets: Inception Score (IS), Fréchet Inception Distance (FID) and Kernel Inception Distance (KID). The best score within two setups are marked in **bold**

| Kernel | Loss | Dimension of Output Layer | CIFAR-10 | | | CelebA | | LSUN | |
|--------|------|--------|------|------|------|------|------|------|------|
| | | | IS | FID | KID $\times 10^3$ | FID | KID $\times 10^3$ | FID | KID $\times 10^3$ |
| Real data | | | 11.28 | 0.217 | 0.00 | 0.589 | 0.00 | 0.670 | 0.00 |
| mix-rbf | mmd | 16 | 6.91 | 28.99 | 20.92 | 22.08 | 14.07 | 68.36 | 41.67 |
| mix-rq | mmd | 16 | 7.12 | 28.23 | 20.13 | 34.02 | 16.96 | 82.08 | 56.41 |
| mix-rbf | mmd | 70 | 6.96 | 27.56 | 19.51 | **21.64** | 14.39 | 65.48 | 40.34 |
| mix-rq | mmd | 70 | **7.19** | 27.68 | 18.88 | 29.78 | **11.91** | 78.89 | 48.82 |
| forest | mmd | 70 | 7.16 | **24.74** | **17.29** | 25.07 | 17.63 | **24.99** | **24.35** |
| rbf-b | mmd-rep | 16 | 7.37 | 23.54 | 17.27 | 27.81 | 15.53 | 17.99 | **16.04** |
| rbf-b | mmd-rep | 70 | **7.60** | 24.80 | 16.38 | 30.82 | 18.36 | 18.28 | 16.57 |
| forest | mmd-rep | 70 | 7.42 | **22.32** | **15.46** | **21.11** | **14.29** | **17.60** | 16.94 |

the discriminator are shown in Appendix A.1. For the sake of comparison with forest kernel, the dimension of discriminator output layer $s$ is set to be 70 for random-forest kernel and to be 16 for other kernels following the previous setting of Bińkowski et al. (2018); Wang et al. (2019).

### 4.1.4 TRAINING HYPER-PARAMETERS

We set the initial learning rate $10^{-4}$ and decrease the learning rate by coefficient 0.8 in iteration 30000, 60000, 90000, and 120000. Adam optimizer (Kingma & Ba, 2015) is used with momentum parameters $\beta_1 = 0.5$ and $\beta_2 = 0.999$. The batch size of each model is 64. All models were trained for 150000 iterations on CIFAR-10, CelebA, and LSUN bedroom datasets, with five discriminator updates per generator update.

### 4.1.5 EVALUATION METRICS

The following three metrics are used for quantitative evaluation: Inception score (IS) (Salimans et al., 2016), Fréchet inception distance (FID) (Heusel et al., 2017), and Kernel inception distance (KID) (Bińkowski et al., 2018). In general, higher IS and Lower FID, KID means better quality. However, outside the dataset Imagenet, the metric IS has some problem, especially for datasets celebA and LSUN bedroom. Therefore, for inception score, we only report the inception score of CIFAR-10. Quantitative scores are calculated based on 50000 generator samples and 50000 real samples.

### 4.2 EXPERIMENTAL RESULTS

We compare our proposed random-forest kernel with mix-rbf kernel and mix-rq kernel in the setting of the MMD GAN loss, and compare our proposed random-forest kernel with rbf-b kernel in the setting with MMD GAN repulsive loss. The Inception Score, the Fréchet Inception Distance and the Kernel Inception Distance of applying different kernels and different loss functions on three benchmark datasets are shown in table 1.

We find that, in the perspective of the original MMD GAN loss, our newly proposed random-forest kernel shows better performance than the mix-rbf kernel and the mix-rq kernel in CIFAR-10 dataset and LSUN bedroom dataset; and in the perspective of the repulsive loss, the performance of our newly proposed random-forest kernel is comparable or better than the rbf-b kernel. The efficacy of our newly proposed random-forest kernel is shown under the setting of both MMD GAN loss and MMD GAN repulsive loss.

Some randomly generated pictures of model learned with various kernels and two different loss functions are visualized in Appendix D.

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

# A SUPPLEMENTARY METHODOLOGY

## A.1 KERNELS AND LOSSES

The formulation of our proposed random-forest kernel is

$$k_{RF}(x, y; \theta_F) = \frac{1}{T}\langle \mu^{(T)}(x; \theta_F), \mu^{(T)}(y; \theta_F)\rangle.$$

In experiments, we take the number of trees in the forest $T = 10$ and the depth of trees $\mathtt{dep} = 3$. Thus, the total number of internal nodes is 70. To notify, in general, the parameters $\theta_F = (\theta_F^t)_{t=1}^T$ are trainable, where $\theta_F^t = (w^t, b^t)$, $w^t = (w_1^t, \ldots, w_{|\mathcal{N}|}^t)$, $b^t = (b_1^t, \ldots, b_{|\mathcal{N}|}^t)$ and $\mathcal{N}$ is the set of every internal nodes of a tree. However, for the sake of experimental simplification, we fix each $w_j^t = 1$, $b_j^t = 0$ untrainable.

In the setting of MMD GAN loss proposed in Li et al. (2017), we compare our random-forest kernel with the two following kernels: Gaussian kernel (Li et al., 2017) with mixture of kernel scales $\sigma$, that is,

$$k_{rbf}^m(x, y) = \sum_{\sigma \in \Sigma} \exp\left(-\frac{\|x-y\|^2}{2\sigma^2}\right),$$

where $\Sigma = \{2, 5, 10, 20, 40, 80\}$, and rational quadratic kernel (Bińkowski et al., 2018) with mixture of kernel scale $\alpha$, that is,

$$k_{rq}^m(x, y) = \sum_{\alpha \in A} \left(1 + \frac{\|x-y\|^2}{2\alpha}\right)^{-\alpha},$$

where $A = \{0.2, 0.5, 1, 2, 5\}$.

In the setting of the MMD GAN with repulsive loss proposed in Wang et al. (2019), we compare our forest kernel with bounded RBF kernel (Wang et al., 2019), that is,

$$k_{rbf}^b(x, y) = \begin{cases} \exp\left(-\dfrac{\max\{\|x-y\|^2, b_l\}}{2\sigma^2}\right) & \text{if } x, y \in \{P_Y\} \\ \exp\left(-\dfrac{\min\{\|x-y\|^2, b_u\}}{2\sigma^2}\right) & \text{if } x, y \in \{P_X\}, \end{cases}$$

with $\sigma = 1$, $b_l = 0.25$ and $b_u = 4$.

## A.2 VISUALIZATION DETAILS OF KERNELS

In figure 3, we compare the contours of three different kernels (the detail of kernels is shown in A.1). We directly plot the filled contours of 2-dimensional Gaussian kernel and rational quadratic kernel with reference to $(0, 0)$; As for random-forest kernel with $T = 10$ and $\mathtt{dep} = 3$, where the input dimension is 70, the steps of a 2-dimensional visualization are as follows:

1) We randomly generate 25000 points from the uniform distribution $\mathcal{U}[-1, 1]^{70}$ and set $(0.5, \ldots, 0.5) \in \mathbb{R}^{70}$ as the reference point. To notify, if the reference point is 70-dimensional zero vector, the values of random-forest kernel will be constant;

2) We calculate the 25000 output values of random-forest kernel;

3) We transform 70-dimensional 25000 randomly generated points and reference point together by t-SNE to 2-dimensional points;

4) We show the filled contour of the neighborhood of transformed reference point. We try to visualize the local structure of random-forest kernel.

## A.3 NETWORK ARCHITECTURE

In the experiments of the CIFAR-10 and LSUN bedroom datasets, we use the DCGAN architecture following Miyato et al. (2018), and for the experiments of the CelebA dataset, we use a 5-layer DCGAN discriminator and a 10-layer ResNet generator as in Bińkowski et al. (2018). The first few layers of the ResNet generator consist of a linear layer and 4 residual blocks as in Gulrajani et al. (2017). The network architecture details are shown in Table 2 and 3.

**Table 2:** Network architecture used in the image generation on `CIFAR-10` dataset and `LSUN` bedroom dataset. In terms of the shape parameter $h$ and $w$ in the table, we take $h = w = 4$ for `CIFAR-10` and $h = w = 8$ for `LSUN` bedroom. As for the output dimension of discriminator $s$, we take $s = 70$ for random-forest kernel and $s = 16, 70$ for other kernels. To notify, 16 is the setting in Bińkowski et al. (2018); Wang et al. (2019) and 70 is set for the comparison of random-forest kernel.

**(a)** Generator

| noise input $z \in \mathbb{R}^{128} \sim \mathcal{U}[-1, 1]$ |
| --- |
| $128 \to h \times w \times 512$, dense, BN, ReLU |
| $4 \times 4$, stride 2 deconv, 256, BN, ReLU |
| $4 \times 4$, stride 2 deconv, 128, BN, ReLU |
| $4 \times 4$, stride 2 deconv, 64, BN, ReLU |
| $3 \times 3$, stride 1 deconv, 3, Sigmoid |

**(b)** Discriminator

| RGB picture $\boldsymbol{x} \in [0, 1]^{H \times W \times 3}$ |
| --- |
| $3 \times 3$, stride 1 conv, 64, LReLU |
| $4 \times 4$, stride 2 conv, 128, LReLU |
| $3 \times 3$, stride 1 conv, 128 LReLU |
| $4 \times 4$, stride 2 conv, 256, LReLU |
| $3 \times 3$, stride 1 conv, 256 LReLU |
| $4 \times 4$, stride 2 conv, 512, LReLU |
| $3 \times 3$, stride 1 conv, 512 LReLU |
| dense $\to s$ |

**Table 3:** Network architecture used in the image generation on `CelebA` dataset. For the output dimension of discriminator $s$, we take $s = 70$ for random-forest kernel and $s = 16, 70$ for other kernels.

**(a)** Generator

| noise input $z \in \mathbb{R}^{128} \sim \mathcal{U}[-1, 1]$ |
| --- |
| $128 \to 5 \times 5 \times 1024$, dense |
| ResBlock up, 512 |
| ResBlock up, 256 |
| ResBlock up, 128 |
| ResBlock up, 64, BN |
| $5 \times 5$, stride 2 deconv, 3, Sigmoid |

**(b)** Discriminator

| RGB picture $\boldsymbol{x} \in [-1, 1]^{H \times W \times 3}$ |
| --- |
| $5 \times 5$, stride 2 conv, 64, LReLU |
| $5 \times 5$, stride 2 conv, 128, LReLU |
| $5 \times 5$, stride 2 conv, 256, LReLU |
| $5 \times 5$, stride 2 conv, 512, LReLU |
| $5 \times 5$, stride 2 conv, 1024, LReLU |
| dense $\to s$ |

# B  THEORETICAL ANALYSIS

In Section B, we will show the main propositions used to prove the Theorem 2. To be specific, in Section B.1, we represent neural networks as computation graphs. In Section B.2, we consider a general class of piecewise analytic functions as the non-linear part of neural networks. In Section B.3, we prove the Lipschitz property of the whole discriminators. In Section B.4, we discover that for $P_X$ almost surely, the network is not differential for its parameters $\theta_N$ and $\psi$. Fortunately, we prove that the measure of bad parameters set is zero.

In Section C, we will show the explicit proof of main propositions in Section B and Theorem 2.

## B.1  NEURAL NETWORKS (NN)

### B.1.1  CONVOLUTIONAL NEURAL NETWORKS (CNN)

Historical attempts to scale up GANs using CNNs to model images have been unsuccessful. The original CNN architecture is made up of convolution, non-linear and pooling. Now for our model, we adopt the deconvolution (Zeiler & Fergus, 2014) net to generate the new data with spatial upsampling. Moreover, batch normalization (Ioffe & Szegedy, 2015) is a regular method which stabilizes learning by normalizing the input to each unit to have zero mean and unit variance. Furthermore, relu functions are used both in generator and discriminator networks as non-linear part. Here we avoid spatial pooling such as max-pooling and global average pooling.

### B.1.2  COMPUTATIONAL GRAPH REPRESENTATION OF NEURAL NETWORKS

Throughout this paper, we always denote by

$$h(\theta) := h(x;\theta) := \{h_1(x;\theta), \ldots, h_{d'}(x;\theta)\}, \tag{5}$$

the output of a fully connected layer, where $d'$ represents the number of neurons in the output.

The general feed-forward networks including CNN and FC can be formulated as a directed acyclic computation graph $\mathcal{G}$ consisting of $L+1$ layers, with a root node $i=0$ and a leaf node $i=L$. For a fixed node $i$, we use the following notations:

$\pi(i)$:  the set of parent nodes of $i$;

$j < i$:  $j$ is a parent node of $i$.

Each node $i > 0$ corresponds to a function $f_i$ that receives a $\mathbb{R}^{d_{\pi(i)}}$-valued input vector, which is the concatenation of the outputs of each layer in $\pi(i)$, and outputs a $\mathbb{R}^{d_i}$-valued vector, where $d_{\pi(i)} = \sum_{j \in \pi(i)} d_j$ and $d_0 = d > 0$. According to the construction of the graph $\mathcal{G}$, the feed-forward network that factorizes with functions $f_i$ recursively can therefore be defined by $h^0 = X$, and for all $0 < i \le L$,

$$h^i := f_i(h^{\pi(i)}),$$

where $h^{\pi(i)}$ is the concatenation of the vectors $h^j$, $j \in \pi(i)$. Here, the functions $f_i$ can be of the following different types:

*(i)* Linear/Affine: If the weights $W^i$ are $m_i$-dimensional vectors, and the function $g_i : \mathbb{R}^{m_i} \to \mathbb{R}^{d_i \times (d_{\pi(i)}+1)}$ is a linear operator on the weights $W^i$, e.g. convolutions, then the functions $f_i$ are of the linear form

$$f_i(Y) = g_i(W^i)\begin{pmatrix} Y & 1 \end{pmatrix}^T =: g_i(W^i)\widetilde{Y}.$$

*(ii)* Non-linear: Such functions $f_i$ including ReLU, max-pooling and ELU, have no learnable weights, can potentially be non-differentiable, and only satisfy some weak regularization conditions, see Definition 3 and the related Examples 2, 3, and 4.

In the following, we denote by $\mathcal{I}$ the set of nodes $i$ such that $f_i$ is non-linear, and its complement by $\mathcal{I}^c := \{1, \ldots, L\} \setminus \mathcal{I}$, that is, the set of all of all linear modules. We write $\theta_N$ as the concatenation of parameters

$$\theta_N := (W^i)_{i \in \mathcal{I}^c} \in \mathbb{R}^{|\theta_N|},$$

where $|\theta_N|$ denotes the total number of parameters of $\theta_N$. Moreover, the feature vector of the network corresponds to the output neurons $\mathcal{G}_L$ of the last layer $L$ and will be denoted by

$$h_\theta := h_{\theta_N}^L \in \mathbb{R}^{|h_{\theta_N}|},$$

where the subscript $\theta$ stands for the parameters of the network. If $X$ is random, we will use $h_{\theta_N}(X)$ to denote explicit dependence on $X$, and otherwise when $X$ is fixed, it will be omitted.

### B.2 PIECEWISE ANALYTIC NONLINEAR LAYERS

Throughout this paper, in the construction of neural networks, we only consider activation functions that are piecewise analytic which can be defined as follows:

**Definition 3 (Piecewise Analytic Functions)** *Let $\{f_i\}_{i \in \mathcal{I}}$ be non-linear layers in neural networks, then $f_i$ is said to be an piecewise analytic function if there exists a partition of $\mathbb{R}^{d_{\pi(i)}}$ with $J_i$ pieces, that is, $\bigcup_{j=1}^{J_i} \bar{\mathcal{D}}_i^j = \mathbb{R}^{d_{\pi(i)}}$ and $\mathcal{D}_i^j \bigcap \mathcal{D}_i^{j'} = \emptyset$ for $j \neq j'$, such that there exist*

(i) *$S_i^j$ real analytic functions $g_{i,s}^j : \mathbb{R}^{d_{\pi(i)}} \to \mathbb{R}$ such that*

$$\mathcal{D}_i^j = \left\{ Y \in \mathbb{R}^{d_{\pi(i)}} \mid g_{i,s}^j(Y) > 0 \text{ for all } s \in \{1, \ldots, S_i^j\} \right\};$$

(ii) *$J_i$ real analytic functions $(f_i^j)_{j \in \{1, \ldots, J_i\}}$ such that*

$$f_i(Y) = f_i^j(Y) \qquad \text{for all } Y \in \bar{\mathcal{D}}_i^j.$$

*The sets $\mathcal{D}_i^j$ are called analytic domains and the functions $f_i^j$ are called analytic value.*

The following examples show that in practice, the vast majority of deep networks satisfy the conditions in the above definition, and therefore are piecewise analytic.

**Example 1 (Sigmoid)** *Let $f_i$ outputs the sigmoid activation function on the inputs. Then we need no partition, and hence there exist*

(i) *$S_i^j = 1$ real analytic functions $g_{i,1}^1(Y) = 1$ such that*

$$\mathcal{D}_i^j = \mathbb{R}^{d_{\pi(i)}}$$

*corresponding to the whole space;*

(ii) *$J_i = 1$ real analytic function $f_i^1 := f_i$.*

Here, we mention that the case in Example 1 corresponds to most of the differentiable activation functions used in deep learning, more examples include the softmax, hyperbolic tangent, and batch normalization functions. On the other hand, in the case that the functions are not differentiable, as is shown in the following examples, many commonly used activation functions including the ReLU activation function are at least piecewise analytic.

**Example 2 (ReLU)** *Let $f_i$ outputs the ReLU activation function on two inputs. Then we have a partition of $J_i = 4$ pieces, with each $\mathcal{D}_i^j$, $j \in \{1, 2, 3, 4\}$, corresponding to a quadrant of the real plane, such that there exist*

(i) *$S_i^j = 2$ real analytic functions $g_{i,1}^j(Y) := \pm Y_1$ and $g_{i,2}^j(Y) := \pm Y_2$ such that*

$$\mathcal{D}_i^j = \{Y \mid g_{i,1}^j(Y) > 0 \text{ and } g_{i,2}^j(Y) > 0\};$$

(ii) *$J_i = 4$ real analytic functions $f_i^1 := (Y_1, Y_2)$, $f_i^2 := (Y_1, 0)$, $f_i^3 := (0, 0)$, and $f_i^4 := (0, Y_2)$ such that*

$$f_i(Y) = f_i^j(Y) \qquad \text{for all } Y \in \mathcal{D}_i^j.$$

Besides the ReLU activation function, other activation functions, such as the Max-Pooling and the ELU (Clevert et al., 2016) also satisfy the form of Definition 3 are therefore piecewise analytic.

**Example 3 (Max-Pooling)** *Let $f_i$ outputs max-pooling on two inputs. Then we have a partition of $J_i = 2$ pieces with each domain $\mathcal{D}_i^j$, $j = 1, 2$, corresponding to a half plane, such that there exist*

(i) *$S_i^1 = S_i^2 = 1$ real analytic function $g_{i,1}^1(Y) := Y_1 - Y_2$ and $g_{i,1}^2(Y) = Y_2 - Y_1$ such that*
$$\mathcal{D}_i^1 = \{Y \mid g_{i,1}^1(Y) > 0\} \qquad and \qquad \mathcal{D}_i^2 = \{Y \mid g_{i,1}^2(Y) > 0\};$$

(ii) *$J_i = 2$ real analytic functions $f_i^1 := Y_1$ and $f_i^2 := Y_2$ such that*
$$f_i(Y) = f_i^j(Y) \qquad for\ all\ Y \in \mathcal{D}_i^j.$$

**Example 4 (ELU)** *Let $f_i$ outputs the ELU activation function on two inputs. Then we have a partition of $J_i = 4$ pieces, with each $\mathcal{D}_i^j$, $j \in \{1, 2, 3, 4\}$, corresponding to a quadrant of the real plane, such that there exist*

(i) *$S_i^j = 2$ real analytic functions $g_{i,1}^j(Y) := \pm Y_1$ and $g_{i,2}^j(Y) := \pm Y_2$ such that*
$$\mathcal{D}_i^j = \{Y \mid g_{i,1}^j(Y) > 0\ and\ g_{i,2}^j(Y) > 0\};$$

(ii) *$J_i = 4$ real analytic functions $f_i^1 := (Y_1, Y_2)$, $f_i^2 := (Y_1, 0)$, $f_i^3 := (0, 0)$, and $f_i^4 := (0, Y_2)$ such that*
$$f_i(Y) = f_i^j(Y) \qquad for\ all\ Y \in \mathcal{D}_i^j.$$

### B.3 BOUNDING THE NETWORK GROWTH

In this section, we investigate the Lipschitz property of the proposed discriminative networks, which is formulated as follows:

Denote $u := (x, y)$, then the linear kernel $k_L(x, y)$ can be written as a univariate function $K_L(u)$, that is,
$$K(u) := K_L(u) := k_L(x, y). \tag{6}$$

**Proposition 4** *Let $\theta_D$ be the parameters of discriminators and $B_r(\theta_D) \subset \mathbb{R}^{|\theta_D|}$ be the ball with center $\theta_D$ and radius $r \in (0, \infty)$. Then, for all $\theta_D' \in B_r(\theta_D)$ and all $x \in \mathbb{R}^d$, there exists a regular function $c(x)$ with $\mathbb{E}_{P_X}[c(x)] < \infty$ such that*
$$\left| K\left(\mu_{\theta_D}^{(T)}(x)\right) - K\left(\mu_{\theta_D'}^{(T)}(x)\right) \right| \le c(x) \|\theta_D - \theta_D'\|.$$

### B.4 EFFECTLESS OF CRITICAL PARAMETERS

In the analysis of section B.3, we intentionally ignore the situation when samples fall into the "bad sets", where the network is not differentiable for data sets $x$ with nonzero measure. And in proposition 5, we show that the measure of the so called "bad sets" is zero.

To better illustrate this proposition, we first denote some notations as follows:

For a fixed $\theta_{N,0} \in \mathbb{R}^{|\theta_N|}$, we denote
$$\mathcal{N}(\theta_{N,0}) = \{x \in \mathbb{R}^d \mid \theta_N \mapsto h_{\theta_N}(x) \text{ is not differentiable at } \theta_{N,0}\} \tag{7}$$
as the set of input vectors $x \in \mathbb{R}^d$ such that $h_{\theta_N}(x)$ is not differentiable with respect to $\theta_N$ at the point $\theta_{N,0}$. Then we call
$$\Theta_{P_X} = \{\theta_N \mid P_X(\mathcal{N}(\theta_N)) > 0\} \tag{8}$$
the set of *critical* parameters, where the network is not differentiable for data sets $x$ with nonzero measure.

**Proposition 5** *Let the set $\Theta_{P_X}$ be as in equation 8. Then, for any distribution $P_X$, we have*
$$\mu(\Theta_{P_X}) = 0.$$

## C   PROOFS

### C.1   PROOFS OF SECTION B.3

To prove Proposition 4, we first introduce Lemma 6 and Lemma 7.

Lemma 6 describes the growth and Lipschitz properties of general networks including convolutional neural networks and fully connected neural networks introduced in Section B.1.

**Lemma 6** *Let $h_{\theta_N}$ be the neural networks defined as in Section B.1. Then there exist continuous functions $a, b : \mathbb{R}^{|\theta_N|} \to \mathbb{R}$ and $\alpha, \beta : \mathbb{R}^{2|\theta_N|} \to \mathbb{R}$ such that for all $x \in \mathbb{R}^d$ and all $\theta_N, \theta'_N \in \mathbb{R}^{|\theta_N|}$, there hold*

$$\|h_{\theta_N}(x)\| \leq b(\theta_N) + a(\theta_N)\|x\|, \tag{9}$$

$$\|h_{\theta_N}(x) - h_{\theta'_N}(x)\| \leq \|\theta_N - \theta'_N\|\big(\beta(\theta_N, \theta'_N) + \alpha(\theta_N, \theta'_N)\|x\|\big). \tag{10}$$

**Proof** [of Lemma 6] We proceed the proof by induction on the nodes of the network.

Obviously, for $i = 0$, the inequalities hold with $b_0 = 0$, $a_0 = 1$, $\beta_0 = 0$ and $\alpha_0 = 0$. Then, for the induction step, let us fix an index $i$. Assume that for all $x \in \mathbb{R}^d$ and all $\theta_N, \theta'_N \in \mathbb{R}^{|\theta_N|}$, there hold

$$\big\|h_{\theta_N}^{\pi(i)}(x)\big\| \leq b_{\pi(i)}(\theta_N) + a_{\pi(i)}(\theta_N)\|x\|,$$

$$\big\|h_{\theta_N}^{\pi(i)}(x) - h_{\theta'_N}^{\pi(i)}(x)\big\| \leq \|\theta_N - \theta_N\|\big(\beta_{\pi(i)}(\theta_N, \theta'_N) + \alpha_{\pi(i)}(\theta_N, \theta'_N)\|x\|\big),$$

where $a_{\pi(i)}(\theta_N)$, $b_{\pi(i)}(\theta_N)$, $\alpha_{\pi(i)}(\theta_N, \theta'_N)$ and $\beta_{\pi(i)}(\theta_N, \theta'_N)$ are some continuous functions.

*(i)* If $i \in \mathcal{I}$, that is, if $i$ is a linear layer, then the following growth condition

$$\|h_{\theta_N}^i\| \leq \big\|g_i(W^i)\tilde{h}_{\theta_N}^{\pi(i)}\big\| \leq \|g_i\|\|W^i\|\big(\big\|h_{\theta_N}^{\pi(i)}\big\| + 1\big) \leq b_i(\theta_N) + a_i(\theta_N)\|x\|$$

holds with $a_i(\theta_N) = \|g_i\|\|W^i\|a_{\pi(i)}(\theta_N)$ and $b_i(\theta_N) = \|g_i\|\|W^i\|b_{\pi(i)}(\theta_N)$. Moreover, concerning with the Lipschitz property, we have

$$\begin{aligned}
\big\|h_{\theta_N}^i(x) - h_{\theta'_N}^i(x)\big\| &= \big\|g_i(W_i)\tilde{h}_{\theta_N}^{\pi(i)}(x) - g_i(W'_i)\tilde{h}_{\theta'_N}^{\pi(i)}(x)\big\| \\
&\leq \big\|g_i(W_i)(\tilde{h}_{\theta'_N}^{\pi(i)}(x) - \tilde{h}_{\theta_N}^{\pi(i)}(x))\big\| + \big\|(g_i(W_i - W'_i))\tilde{h}_{\theta'_N}^{\pi(i)}(x)\big\| \\
&\leq \|g_i\|\big(\|W_i\|\big\|h_{\theta'_N}^{\pi(i)}(x) - h_{\theta_N}^{\pi(i)}(x)\big\| + \|W_i - W'_i\|\big(\big\|h_{\theta'_N}^{\pi(i)}(x)\big\| + 1\big)\big) \\
&\leq \|\theta_N - \theta'_N\|\big(\beta_i(\theta_N, \theta'_N) + \alpha_i(\theta_N, \theta'_N)\|x\|\big),
\end{aligned}$$

where we used the notations

$$\alpha_i(\theta_N, \theta'_N) := \|g_i\|\big((\|W_i\| + \|W'_i\|)\alpha_{\pi(i)}(\theta_N, \theta'_N) + (a_{\pi(i)}(\theta_N) + a_{\pi(i)}(\theta'_N))\big)$$

$$\beta_i(\theta_N, \theta'_N) := \|g_i\|\big((\|W_i\| + \|W'_i\|)\beta_{\pi(i)}(\theta_N, \theta'_N) + (b_{\pi(i)}(\theta_N) + b_{\pi(i)}(\theta'_N)) + 1\big).$$

*(ii)* If $i \in \mathcal{I}^c$, that is, $i$ is not a linear layer, here we only consider the sigmoid and ReLU functions. We first show that both of them are Lipschitz continuous. Concerning the sigmoid function,

$$\sigma(x) = \frac{1}{1 + e^{-x}}$$

we obviously have

$$\sigma'(x) = \frac{1}{2 + e^{-x} + e^x} = \frac{1}{(e^{-x/2} + e^{x/2})^2} \leq \frac{1}{4},$$

for all $x \in \mathbb{R}$. Consequently, the sigmoid function is Lipschitz continuous with Lipschitz constant $|\sigma|_1 := 1/4$. Next, for the ReLU function,

$$\sigma(x) = \max\{0, x\},$$

we have for all $x \in \mathbb{R}$,

$$\sigma'(x) = \max\{0, 1\} \leq 1.$$

Therefore, the ReLU function is Lipschitz continuous with Lipschitz constant $|\sigma|_1 := 1$. Thus, non-linear layer $f_i$ is Lipschitz continuous with Lipschitz constant $M := |f_i|_1$.

Consequently, by recursion, we obtain continuous functions $\alpha_i = M\alpha_{\pi(i)}$, $\beta_i = M\beta_{\pi(i)}$, $a_i = Ma_{\pi(i)}$, and $b_i = Mb_{\pi(i)}$. ∎

Next, we investigate the growth conditions and Lipschitz property of the random forest. For the ease of notations, we write the function $\mu^{(T)}$ as

$$\mu^{(T)}(x; \theta_N, \theta_F) =: \mu^{(T)}(h(x; \theta_N); \theta_F) =: \mu^{(T)}(h).$$

Moreover, it is easily seen that $T \cdot |\mathcal{N}|$ equals the number of internal nodes in the random forest.

**Lemma 7** *Let $h(x; \theta_N)$ be the input vector of random trees and $\theta_F := (w, b)$ where $w$ and $b$ denote the weights and bias of the random forests, respectively. Then, for all $h \in \mathbb{R}^{d'}$ and $\theta_F, \theta'_F \in \mathbb{R}^{2T|\mathcal{N}|}$, there exist continuous functions $c_1$, $c_2$, and constants $c_3$, $c_4$, $c_5$ such that*

$$\left\| \mu^{(T)}(h(x; \theta_N); \theta_F) \right\| \leq c_1(b) + c_2(w)\|h(x; \theta_N)\|; \tag{11}$$

$$\left\| \mu^{(T)}(h(x; \theta_N); \theta_F) - \mu^{(T)}(h(x; \theta_N); \theta'_F) \right\| \leq c_3\|w - w'\|\|h\| + c_4\|b - b'\|; \tag{12}$$

$$\left\| \mu^{(T)}(h(x; \theta_N); \theta_F) - \mu^{(T)}(h(x; \theta'_N); \theta_F) \right\| \leq c_5\|w\|\|h(x; \theta_N) - h(x; \theta'_N)\|. \tag{13}$$

**Proof** [of Lemma 7] For $t = 1, \ldots, T$, denote $\mu_\ell^t$ as the $\ell$-th leaf node of the $t$-th tree. By convenient abuse of notation, we write the function $\mu_\ell^t$ as

$$\mu_\ell^t(x; \theta_N, w^t, b^t) =: \mu_\ell^t(h(x; \theta_N); w^t, b^t) =: \mu_\ell(h).$$

First of all, let us verify the growth condition equation 11. From the definition of the function $T$ we immediately know that for any node $i$ in the random forest, there exists an element $j = T(i)$ such that

$$p_i(h) = \frac{1}{1 + \exp^{-(w_i h_j + b_i)}}.$$

Obviously, we have

$$\max_i \{p_i, 1 - p_i\} \leq \frac{1}{1 + \exp^{-|w_i h_j + b_i|}}.$$

For any leaf node $\ell \in \mathcal{L}_t$, that is, for any node in the $t$-th tree, in the following, for the ease of notation, we always set $\mu_\ell = \mu_\ell^t$, $w_i = w_i^t$, $h_j = h_j^t$, $b_i = b_i^t$, if not otherwise mentioned. Then $\mu_\ell$ in equation 3 can be upper bounded by

$$\mu_\ell \leq \frac{1}{\prod_{i \in \mathcal{R}_\ell^t} \left(1 + e^{-|w_i h_j + b_i|}\right)} \leq \frac{1}{2} + \sum_{i \in \mathcal{R}_\ell^t} |w_i h_j + b_i|$$

$$\leq \frac{1}{2} + \sum_{i=1}^{|\mathcal{N}|} |w_i h_j + b_i| \leq \frac{1}{2} + \sum_{i=1}^{|\mathcal{N}|} (|w_i h_j| + |b_i|).$$

Taking the summation on both sides of the above inequality with respect to all of the nodes in the random forest, that is, w.r.t. $\ell \in \mathcal{L}$, we obtain

$$\left\| \mu^{(T)} \right\| \leq \frac{|\mathcal{L}|}{2} + |\mathcal{N}| \sum_{i,t} |b_i^t| + |\mathcal{N}| \sum_{i,t} |w_i^t| |h_{T(i)}|$$

$$\leq \frac{|\mathcal{L}|}{2} + |\mathcal{N}||\mathcal{L}|\|b\| + |\mathcal{N}| \left( \sum_{i,t} |w_i^t|^2 \right)^{1/2} \left( \sum_{i,t} |h_{T(i)}|^2 \right)^{1/2}$$

$$\leq \frac{|\mathcal{L}|}{2} + |\mathcal{N}||\mathcal{L}|\|b\| + |\mathcal{N}|\|w\| \frac{|\mathcal{L}|}{d'} \|h\|$$

$$=: c_1(b) + c_2(w)\|h\|,$$

where the second inequality follows from the Cauchy-Schwartz inequality and the third inequality is due to the fact that the number that the nodes $p_i$ in the random forest assigned to the corresponding node $h_j$ equals $\lfloor T|\mathcal{N}|/d' \rfloor$ or $\lfloor T|\mathcal{N}|/d' \rfloor + 1$, which are less than $|\mathcal{L}|/d'$.

Now, we show the Lipschitz properties equation 12 and equation 13 of the random forest. From the equivalent form equation 3 concerning the value of the leaf node, we easily see that $\mu_\ell$ can be written as a product of probability functions $p_i$ or $1 - p_i$. Therefore, without loss of generality, we can assume $\mu_\ell$ are of the product form

$$\mu_\ell := p_{\mathcal{R}_\ell(1)} \cdot \ldots \cdot p_{\mathcal{R}_\ell(\mathcal{H})}. \tag{14}$$

For a fixed $t = 1, \ldots, T$, recall that $T_t$ denotes the connection function of the $t$-th random tree. Then, the Lipschitz property of the sigmoid function and the continuously differentiability of the linear transform yield

$$|p_i - p_i'| \leq |p_i(h; \theta_F) - p_i(h; \theta_F')| \leq |w_i h_i + b_i - w_i' h_i - b_i'| \leq |w_i - w_i'||h_i| + |b_i - b_i'|. \tag{15}$$

Then, equation 14 together with equation 15 implies

$$\begin{aligned}
|\mu_\ell^t(h; \theta_F) - \mu_\ell^t(h; \theta_F')| &= |p_{\mathcal{R}_\ell(1)} \times \cdots \times p_{\mathcal{R}_\ell(\mathcal{H})} - p_{\mathcal{R}_\ell(1)}' \times \cdots \times p_{\mathcal{R}_\ell(\mathcal{H})}'| \\
&\leq |p_{\mathcal{R}_\ell(1)} - p_{\mathcal{R}_\ell(1)}'| \times p_{\mathcal{R}_\ell(2)} \times \cdots \times p_{\mathcal{R}_\ell(\mathcal{H})} \\
&\quad + \cdots + p_{\mathcal{R}_\ell(1)}' \times \cdots \times p_{\mathcal{R}_\ell(\mathcal{H}-1)}' \times |p_{\mathcal{R}_\ell(\mathcal{H})} - p_{\mathcal{R}_\ell(\mathcal{H})}'| \\
&\leq \sum_{i=1}^{\mathcal{H}} |p_{\mathcal{R}_\ell(i)} - p_{\mathcal{R}_\ell(i)}'| \leq \sum_{k=1}^{|\mathcal{N}|} |p_k - p_k'| \\
&\leq \sum_{k=1}^{|\mathcal{N}|} |w_k^t - w_k'||h_i| + |b_k^t - b_k'| \\
&\leq \|w^t - w^{t'}\| \left( \sum_{k=1}^{|\mathcal{N}|} h_{T_t(k)}^2 \right)^{1/2} + |\mathcal{N}| \|b^t - b^{t'}\|,
\end{aligned}$$

where the last inequality follows from the Cauchy-Schwartz inequality. Consequently, concerning the random forest, we obtain

$$\begin{aligned}
\|\mu^{(T)}(h; \theta_F) - \mu^{(T)}(h; \theta_F')\| &\leq |\mathcal{N}| \sum_t \|w^t - w^{t'}\| \left( \sum_{k=1}^{|\mathcal{N}|} h_{T_t(k)}^2 \right)^{1/2} + |\mathcal{N}|^2 \sum_t \|b^t - b^{t'}\| \\
&\leq |\mathcal{N}| \|w - w'\| \left( \sum_{k,t} h_{T_t(k)}^2 \right)^{1/2} + |\mathcal{N}|^2 \sqrt{T} \|b - b'\| \\
&\leq |\mathcal{N}| \|w - w'\| \frac{\mathcal{L}}{d'} \|h\| + |\mathcal{N}||\mathcal{L}|\|b - b'\| \\
&=: c_3 \|w - w'\|\|h\| + c_4 \|b - b'\|,
\end{aligned}$$

where the second inequality is again due to then Cauchy-Schwartz inequality.

Analogously, for a fixed $t = 1, \ldots, T$ and any $i$, we have

$$|p_i(h; \theta_F) - p_i(h'; \theta_F)| \leq |w_i h_j' + b_i - w_i h_j' - b_i| \leq |w_i||h_j - h_j'|,$$

and consequently we obtain

$$
\begin{aligned}
\mu_\ell^t(h;\theta_F) - \mu_\ell^t(h';\theta_F) &= |p_{\mathcal{R}_\ell(1)} \times \cdots \times p_{\mathcal{R}_\ell(\mathcal{H})} - p'_{\mathcal{R}_\ell(1)} \times \cdots \times p'_{\mathcal{R}_\ell(\mathcal{H})}| \\
&\leq |p_{\mathcal{R}_\ell(1)} - p'_{\mathcal{R}_\ell(1)}| \times p_{\mathcal{R}_\ell(2)} \times \cdots \times p_{\mathcal{R}_\ell(\mathcal{H})} \\
&\quad + \cdots + p'_{\mathcal{R}_\ell(1)} \times \cdots \times p'_{\mathcal{R}_\ell(\mathcal{H}-1)} \times |p_{\mathcal{R}_\ell(\mathcal{H})} - p'_{\mathcal{R}_\ell(\mathcal{H})}| \\
&\leq \sum_{j=1}^{\mathcal{H}} |p_{\mathcal{R}_\ell(j)} - p'_{\mathcal{R}_\ell(j)}| \leq \sum_{i=1}^{|\mathcal{N}|} |p_i^t - p_i^{t'}| \\
&\leq \sum_{i=1}^{|\mathcal{N}|} |w_i^t| |h_{T_t(i)} - h'_{T_t(i)}| \\
&\leq \|w^t\| \left( \sum_{i=1}^{|\mathcal{N}|} |h_{T_t(i)} - h'_{T_t(i)}|^2 \right)^{1/2}
\end{aligned}
$$

and for the random forest, there holds

$$
\begin{aligned}
\left\| \mu^{(T)}(h;\theta_F) - \mu^{(T)}(h';\theta_F) \right\| &\leq |\mathcal{N}| \sum_t \|w^t\| \left( \sum_{i=1}^{|\mathcal{N}|} |h_{T_t(i)} - h'_{T_t(i)}|^2 \right)^{1/2} \\
&\leq |\mathcal{N}| \|w\| \left( \sum_{i,t} |h_{T_t(i)} - h'_{T_t(i)}|^2 \right)^{1/2} \\
&\leq |\mathcal{N}| \|w\| \frac{\mathcal{L}}{d'} \|h - h'\| \\
&=: c_5 \|w\| \|h - h'\|,
\end{aligned}
$$

which completes the proof. ∎

The next proposition presents the growth condition and the Lipschitz property of the composition of the neural network and the random forest.

**Lemma 8** *Let $B_r(\theta_D) \subset \mathbb{R}^{|\theta_D|}$ be the ball with center $\theta_D$ and radius $r \in (0,\infty)$. Then, for all $\theta'_D \in B_r(\theta_D)$, all $x \in \mathbb{R}^d$, there exist continuous functions $c_6$, $c_7$, $c_8$, and $c_9$ such that*

$$
\left\| \mu^{(T)}(x;\theta_D) \right\| \leq c_7(\theta_D) + c_6(\theta_D)\|x\|;
$$
$$
\left\| \mu^{(T)}(x;\theta_D) - \mu^{(T)}(x;\theta'_D) \right\| \leq \|\theta_D - \theta'_D\|\big(c_8(\theta_D,\theta'_D) + c_9(\theta_D,\theta'_D)\|x\|\big).
$$

**Proof** [of Lemma 8] Combining equation 9 in Lemma 6 with equation 11 in Lemma 7, we obtain the growth condition of the form

$$
\begin{aligned}
\left\| \mu^{(T)}(x;\theta_N,\theta_F) \right\| &\leq c_1(b) + c_2(w)\|h(x;\theta_N)\| \\
&\leq c_1(b) + c_2(w)(b(\theta_N) + a(\theta_N)\|x\|) \\
&= c_1(b) + c_2(w)b(\theta_N) + c_2(w)a(\theta_N)\|x\| \\
&=: c_7(b,w,\theta_N) + c_6(w,\theta_N)\|x\| \\
&= c_7(\theta_D) + c_6(\theta_D)\|x\|.
\end{aligned}
$$

Concerning the Lipschitz property, using equation 12 and equation 13, we get

$$
\begin{aligned}
&\left\| \mu^{(T)}(x;\theta_N,\theta_F) - \mu^{(T)}(x;\theta'_N,\theta'_F) \right\| \\
&\leq \left\| \mu^{(T)}(x;\theta_N,\theta_F) - \mu^{(T)}(x;\theta'_N,\theta_F) \right\| + \left\| \mu^{(T)}(x;\theta'_N,\theta_F) - \mu^{(T)}(x;\theta'_N,w',b') \right\| \\
&\leq c_5\|w\|\|h(x;\theta_N) - h(x;\theta'_N)\| + c_3\|w - w'\|\|h(x;\theta'_N)\| + c_4\|b - b'\| \\
&\leq c_5\|w\|\|\theta_N - \theta'_N\|(\beta(\theta_N,\theta'_N) + \alpha(\theta_N,\theta'_N)\|x\|) \\
&\quad + c_3\|w - w'\|(b(\theta_N) + a(\theta_N)\|x\|) + c_4\|b - b'\|,
\end{aligned}
$$

where the last inequality follows from equation 9 and equation 10 established in Proposition 6. With the concatenation $\theta_D := (\theta_N, \theta_F) := (\theta_N, w, b)$ we obtain

$$\left\| \mu^{(T)}(x; \theta_D) - \mu^{(T)}(x; \theta'_D) \right\| \leq \|\theta_D - \theta'_D\| (c_8(\theta_D, \theta'_D) + c_9(\theta_D, \theta'_D) \|x\|)$$

and thus the assertion is proved. ∎

**Proof** [of Proposition 4] First we give the growth conditions for the linear kernel. Let $k$ be the linear kernel $k_L(x, y) := \langle x, y \rangle$, then we have

$$|k_L(x, y)| \leq \|x\|^2 + \|y\|^2 + 1,$$

$$\|\nabla_{x,y} k_L(x, y)\| \leq \left( \|x\|^2 + \|y\|^2 \right)^{1/2} + 1.$$

If we denote $u := (x, y)$, then the above linear kernel $k_L(x, y)$ can be written a as a univariate function $K_L(u)$, that is,

$$K(u) := K_L(u) := k_L(x, y). \tag{16}$$

Therefore we have $K$ is continuously differentiable satisfying the following growth conditions:

$$|K(u)| \leq \|u\|^2 + 1, \tag{17}$$

$$\|\nabla_u K(u)\| \leq \|u\| + 1. \tag{18}$$

Let us define the function $f : [0, 1] \to \mathbb{R}$ by

$$f(t) = K(tu + (1 - t)v), \qquad t \in [0, 1].$$

Then we have $f(0) = K(v)$ and $f(1) = K(u)$. Moreover, $f$ is differentiable with derivative

$$f'(t) = \nabla K((tu + (1 - t)v))(u - v).$$

The growth condition equation 18 implies

$$\begin{aligned}
|f'(t)| &= \|\nabla K((tu + (1 - t)v)(u - v)\| \\
&\leq \|\nabla K(tu + (1 - t)v)\| \|(u - v)\| \\
&\leq \left( \|t(u - v) + v\| + 1 \right) \|u - v\| \\
&\leq \left( (\|u - v\| + \|v\|) + 1 \right) \|u - v\|.
\end{aligned}$$

Using the mean value theorem, we obtain that for all $u, v \in \mathbb{R}^L$, there holds

$$|K(u) - K(v)| \leq \left( (\|u - v\| + \|v\|) + 1 \right) \|u - v\|.$$

With $u := \mu^{(T)}_{\theta'_D}(x)$ and $v := \mu^{(T)}_{\theta_D}(x)$, one gets

$$\left| K\big(\mu^{(T)}_{\theta'_D}(x)\big) - K\big(\mu^{(T)}_{\theta_D}(x)\big) \right| \leq \left( \|\mu^{(T)}_{\theta'_D}(x) - \mu^{(T)}_{\theta_D}(x)\| + \|\mu^{(T)}_{\theta_D}(x)\| + 1 \right) \|\mu^{(T)}_{\theta'_D}(x) - \mu^{(T)}_{\theta_D}(x)\|.$$

Proposition 8 tells us that

$$\left\| \mu^{(T)}_{\theta'_D}(x) - \mu^{(T)}_{\theta_D}(x) \right\| \leq \left( c_8(\theta_D, \theta'_D) + c_9(\theta_D, \theta'_D) \|x\| \right) \|\theta_D - \theta'_D\|,$$

$$\left\| \mu^{(T)}_{\theta_D}(x) \right\| \leq c_7(\theta_D) + c_6(\theta_D) \|x\|.$$

holds for some continuous functions $c_6$, $c_7$, $c_8$, and $c_9$, which are also bounded by certain constant $B > 0$ on the ball $B_r(\theta_D)$. Some elementary algebra shows that

$$\left| K\big(\mu^{(T)}_{\theta'_D}(x)\big) - K\big(\mu^{(T)}_{\theta_D}(x)\big) \right| \leq \left( B^2(r + 1)(1 + \|x\|)^2 + B(1 + \|x\|) \right) \|\theta'_D - \theta_D\|.$$

Since $t \mapsto (1 + t^{1/2})^2$ is concave on $t \geq 0$, Jensen's inequality together with the moment assumption $\mathbb{E}\|x\|^2 < \infty$ implies

$$\mathbb{E}(1 + \|x\|)^2 \leq \left( 1 + (\mathbb{E}\|x\|^2)^{1/2} \right)^2 < \infty$$

and also $\mathbb{E}(1 + \|x\|) < \infty$ by the moment assumption. Therefore, the regular function $c(x)$ defined by

$$c(x) := \left( B^2(r + 1)(1 + \|x\|)^2 + B(1 + \|x\|) \right)$$

is integrable and thus our assertion is proved. ∎

## C.2 Proofs of Section B.4

To reach the proof of Proposition 5, we first introduce Lemma 9 and Lemma 11.

Let $i$ be a fixed node. To describe *paths* through the network's computational graph, we need to introduce the following notations:

$$\mathcal{P} := \{(i,j,s) \in \mathbb{N}^3 \mid i \in \{0,1,\ldots,L\}, j \in \{1,\ldots,J_i\}, s \in \{1,\ldots,S_{i,j}\}\};$$

$$\mathcal{A}(i) := \{i' \mid i' \text{ is an ancestor of } i\};$$

$$\neg i := \{(i',j,s) \in \mathcal{P} \mid i' \in \mathcal{A}(i) \text{ or } i' = i\};$$

$$\neg \pi(i) := \bigcup_{i' \in \pi(i)} \neg i' := \{(i',j,s) \in \mathcal{P} \mid i' \in \mathcal{A}(i)\};$$

$$\partial i := \{(i,j,s) \in \mathcal{P}\}.$$

Obviously, we always have $\partial i \subseteq \neg i$ and $\neg i = \partial i \cup \neg \pi(i)$.

We define a *backward trajectory* starting from node $i$ by

$$q := (i', j_{i'})_{i' \in \mathcal{A}(i) \cup \{i\}}, \qquad j_{i'} \in \{1,\ldots,J_{i'}\}.$$

The set of all backward trajectories for node $i$ will be denoted by $\mathcal{Q}(i)$.

**Lemma 9** *Let $i$ be a fixed node in the network graph. If $\theta_N \in \mathbb{R}^{|\theta_N|} \setminus \partial \mathcal{S}^{\neg i}$, then there exist a constant $\eta > 0$ and a trajectory $q \in \mathcal{Q}(i)$ such that for all $\theta'_N \in B_\eta(\theta_N)$, there holds*

$$h^i_{\theta'_N} = f^q(\theta'_N),$$

*where $f^q$ is the real analytic function on $\mathbb{R}^{|\theta_N|}$ with the same structure as $h_{\theta_N}$, only replacing each nonlinear $f_{i'}$ with the analytic function $f^{j_{i'}}_{i'}$ for $(i', j_{i'}) \in q$.*

**Proof** [of Lemma 9] We proceed by induction on the nodes of the network. If $i = 0$, we obviously have $h^0_{\theta_N} = x$, which is real analytic on $\mathbb{R}^{|\theta_N|}$. For the induction step we assume that the assertion holds for $\neg \pi(i)$ and let $\theta_N \in \mathbb{R}^{|\theta_N|} \setminus \partial \mathcal{S}^{\neg i}$. Then there exist a constant $\eta > 0$ and a trajectory $q \in \mathcal{Q}(\pi(i))$ such that for all $\theta'_N \in B_\eta(\theta_N)$, there holds

$$h^{\pi(i)}_{\theta'_N} = f^q(\theta'_N) \tag{19}$$

with $f^q : \mathbb{R}^{|\theta_N|} \to \mathbb{R}$ being a real analytic function.

*(i)* If $\theta_N \notin \mathcal{S}^{\partial i}$, then there exists a sufficiently small constant $\eta' > 0$ such that $B_{\eta'}(\theta_N) \cap \mathcal{S}^{\partial i} = \emptyset$. Therefore, there exists a $j \in \{1,\ldots,J_i\}$ such that for all $\theta'_N \in B_{\eta'}(\theta_N)$, there holds

$$h^i_{\theta'_N} = f^j_i(h^{\pi(i)}_{\theta'_N})$$

where $f^j_i$ is one of the real analytic functions in the definition of $f_i$. Then, equation 19 implies that for all $\theta'_N \in B_{\min\{\eta,\eta'\}}(\theta_N)$, there holds

$$h^i_{\theta'_N} = f^j_i(f^q(\theta'_N)). \tag{20}$$

*(ii)* Consider the case $\theta_N \in \mathcal{S}^{\partial i}$. By assumption, we have $\theta_N \notin \partial \mathcal{S}^{\partial i}$. Then there exists a small enough constant $\eta' > 0$ such that $B_{\eta'}(\theta_N) \subset \mathcal{S}^{\partial i}$. If we denote

$$A := \{p = (i,j,s) \in \partial i \mid \theta_N \in \mathcal{S}^{\mathcal{P}}\},$$

then $A \neq \emptyset$ since $\theta_N \in \mathcal{S}^{\partial i}$. Therefore, we have

$$\theta_N \in \bigcap_{p \in A} \mathcal{S}^p \qquad \text{and} \qquad \theta_N \notin \bigcup_{p \in A^c} \mathcal{S}^p.$$

Now we show by contradiction that for $\eta'$ small enough, there holds

$$B_{\eta'}(\theta_N) \subseteq \bigcap_{p \in A} \mathcal{S}^p.$$

To this end, we assume that there exists a sequence of parameter and index pairs $(\theta_{N,n}, p_n)$ such that $p_n \in A^c$, $\theta_{N,n} \in \mathcal{S}^{p_n}$, and $\theta_{N,n} \to \theta_N$ as $n \to \infty$. Since $A^c$ is finite, there exists a constant subsequence $\{p_{n_i}\} \subset \{p_n\}$ and some constant $p_0 \in A^c$ with $p_{n_i} = p_0$ for all $i$. Then the continuity of the network and $g_{p_0}$ imply that $\mathcal{S}^{p_0}$ is a closed set and consequently we obtain $\theta_N \in \mathcal{S}^{p_0}$ by taking the limit, which contradicts the fact that $\theta_N \notin \bigcup_{p \in A^c} \mathcal{S}^p$. Therefore, for $\eta'$ small enough, there holds $B_{\eta'}(\theta_N) \subseteq \bigcap_{p \in A} \mathcal{S}^p$, which contradicts the assumption $\theta_N \notin \partial \mathcal{S}^{\partial i}$. Consequently, there exists a $j \in \{1, \ldots, J_i\}$ satisfying equation 20. By setting $f^{q_0} = f_i^j(f^q)$ with $q_0 = ((i,j) \oplus q) \in \mathcal{Q}(i)$, where $\oplus$ denotes concatenation, then for for all $\theta'_N \in B_{\min\{\eta, \eta'\}}(\theta_N)$, there holds

$$h_{\theta'_N}^i = f^{q_0}(\theta'_N)$$

and the assertion is proved. ∎

Now, for a fixed $p = (i, j, s) \in \mathcal{P}$, we denote the set of network parameters $\theta_N$ that lie on the boundary of $p$ by

$$\mathcal{S}^p = \{\theta_N \in \mathbb{R}^{|\theta_N|} \mid g_{i,s}^j(h_{\theta_N}^{\pi(i)}) = 0\},$$

where the functions $g_{i,s}^j$ are as in Definition 3. As usual, the boundary of the set $\mathcal{S}^p$ is denoted by $\partial \mathcal{S}^p$ and the set of the boundaries is denoted by

$$\partial \mathcal{S}^{\mathcal{P}} := \bigcup_{p \in \mathcal{P}} \partial \mathcal{S}^p. \tag{21}$$

Finally, for the ease of notations, if $\mathcal{P}' \subset \mathcal{P}$, we write

$$\mathcal{S}^{\mathcal{P}'} := \bigcup_{p' \in \mathcal{P}'} \mathcal{S}^{p'}, \qquad \partial \mathcal{S}^{\mathcal{P}'} := \bigcup_{p' \in \mathcal{P}'} \partial \mathcal{S}^{p'}. \tag{22}$$

Obviously, we have $\theta_N \in \mathbb{R}^m \setminus \partial \mathcal{S}^{\neg \pi(i)}$. To prove Lemma 11, we need the following lemma which follows directly from Mityagin (2015) and hence we omit the proof.

**Lemma 10** *Let $\theta_N \mapsto F(\theta_N) : \mathbb{R}^{|\theta_N|} \to \mathbb{R}$ be a real analytic function and define*

$$\mathcal{M} := \{\theta_N \in \mathbb{R}^{|\theta_N|} \mid F(\theta_N) = 0\}.$$

*Then we have either $\mu(\mathcal{M}) = 0$ or $F = 0$.*

**Lemma 11** *Let the set of boundaries $\partial \mathcal{S}^{\mathcal{P}}$ be as in equation 21. Then we have*

$$\mu(\partial \mathcal{S}^{\mathcal{P}}) = 0.$$

**Proof** [of Lemma 11] We proceed the proof by induction. For $i = 0$, we obviously have $\partial \mathcal{S}^{\neg 0} = \emptyset$ and therefore $\mu(\partial \mathcal{S}^{\neg 0}) = 0$. For the induction step let us assume that

$$\mu(\partial \mathcal{S}^{\neg \pi(i)}) = 0.$$

For $s = (p, q)$, the pair of an index $p \in \partial i$ and a trajectory $q \in \mathcal{Q}(i)$, we define

$$\mathcal{M}_s := \{\theta_N \in \mathbb{R}^{|\theta_N|} \mid g_p(f^q(\theta_N)) = 0\},$$

where the analytic function $f^q$ is defined as in Proposition 9. Then we prove by contradiction that for any $\theta_N \in \partial \mathcal{S}^{\partial i} \setminus \partial \mathcal{S}^{\neg \pi(i)}$, there exists an $s \in \partial i \times \mathcal{Q}(i)$ such that $\theta_N \in \mathcal{M}_s$ and

$$\mu(\mathcal{M}_s) = 0.$$

To this end, let $\theta_N \in \partial \mathcal{S}^{\partial i} \setminus \partial \mathcal{S}^{\neg \pi(i)}$, then for small enough $\eta > 0$, there holds $B_\eta(\theta_N) \subset \mathbb{R}^{|\theta_N|} \setminus \partial \mathcal{S}^{\neg \pi(i)}$. By Proposition 9, there exists a trajectory $q \in \mathcal{Q}(\pi(i))$ such that for all $\theta'_N \in B_\eta(\theta_N)$, there holds

$$h_{\theta'_N}^{\pi(i)} = f^q(\theta'_N). \tag{23}$$

Moreover, since $\theta_N \in \partial \mathcal{S}^{\partial i}$, there exists an index $p \in \partial i$ such that $g_p(h_{\theta_N}^{\pi(i)}) = 0$. This means that for $s = (p, q)$, we have $\theta_N \in \mathcal{M}_s$. Therefore, we have

$$\partial \mathcal{S}^{\partial i} \setminus \partial \mathcal{S}^{\neg \pi(i)} \subset \bigcup_{s \in A} \mathcal{M}_s, \tag{24}$$

where $A \subseteq \mathcal{P} \times \bigcup_{j=0}^{L} \mathcal{Q}(j)$ is finite. Suppose that $\mu(\mathcal{M}_s) > 0$, then by Lemma 10, we have $\mathcal{M}_s = \mathbb{R}^{|\theta_N|}$ and hence $B_\eta(\theta_N) \subset \mathcal{M}_s$. By equation 23, we then have $B_\eta(\theta_N) \subset \mathcal{S}^{\partial i}$, which contradicts the fact that $\theta_N \in \partial \mathcal{S}^{\partial i}$ and hence we have

$$\mu(\mathcal{M}_s) = 0. \tag{25}$$

Combing equation 24 and equation 25, we obtain

$$\mu\big(\partial \mathcal{S}^{\partial i} \setminus \partial \mathcal{S}^{\neg \pi(i)}\big) \leq \sum_{s \in A} \mu(\mathcal{M}_s) = 0.$$

By the assumption $\mu(\partial \mathcal{S}^{\neg \pi(i)}) = 0$, we conclude that $\mu(\partial \mathcal{S}^{\neg i}) = 0$. Since for the last node $L$, we have $\neg L = \mathcal{P}$ and therefore $\mu(\partial \mathcal{S}^P) = 0$. ∎

Note that the random forest $\mu^{(T)}(\cdot)$ can be considered as the composition of affine transformations and sigmoid functions and hence are always continuously differentiable with respect to $\theta_F$, we only need to investigate whether the neural network $h_{\theta_N}(x)$ is differentiable with respect to $\theta_N$. For a fixed $x \in \mathbb{R}^d$, we write

$$\Theta_x = \{\theta_{N,0} \in \mathbb{R}^{|\theta_N|} \mid \theta_N \mapsto h_{\theta_N}(x) \text{ is not differentiable at } \theta_{N,0}\} \tag{26}$$

as the set of parameters for which the network is not differentiable.

**Lemma 12** *Let the set $\Theta_x$ be as in equation 26. Then, for any $x \in \mathbb{R}^d$, we have*

$$\mu(\Theta_x) = 0.$$

**Proof** [of Lemma 12] Let the boundary set $\partial \mathcal{S}^P$ be defined as in equation 22 and $\theta_{N,0} \in \Theta_x$. Obviously, we have $\theta_0 \in \mathcal{S}^P$. We proceed the proof of the inclusion $\Theta_x \subseteq \partial \mathcal{S}^P$ by contradiction. To this end, we assume that $\theta_{D,0} \notin \partial \mathcal{S}^P$. When Lemma 9 applied to the output layer, there exist some $\eta > 0$ and a real analytic function $f(\theta_N)$ such that for all $\theta_N \in B_\eta(\theta_{N,0})$, there holds

$$h_{\theta_N} = f(\theta_N).$$

Consequently, the network is differentiable at $\theta_{N,0}$, contradicting the fact that $\theta_{N,0} \in \Theta_x$. Therefore, we have $\Theta_x \subseteq \partial \mathcal{S}^P$ and hence $\mu(\Theta_x) = 0$ since $\mu(\partial \mathcal{S}^P) = 0$ according to Lemma 11. ∎

**Proof** [of Proposition 5] Let the sets $\mathcal{N}(\theta_N)$, $\Theta_{P_X}$, and $\Theta_x$ be as in equation 7, equation 8, and equation 26, respectively. Consider the sets

$$S_1 = \{(\theta_N, x) \in \mathbb{R}^{|\theta_N|} \times \mathbb{R}^d \mid \theta_N \in \Theta_{P_X} \text{ and } x \in \mathcal{N}(\theta_N)\},$$
$$S_2 = \{(\theta_N, x) \in \mathbb{R}^{|\theta_N|} \times \mathbb{R}^d \mid \theta_N \in \Theta_x\}.$$

Since the network is continuous and not differentiable, Theorem I in Piranian (1966) implies that the sets $S_1$ and $S_2$ are measurable. Obviously, we have $S_1 \subset S_2$ and therefore we obtain $\nu(S_1) \leq \nu(S_2)$, where $\nu := P_X \otimes \mu$. On the one hand, Fubini's theorem implies

$$\nu(S_2) = \int_{\mathbb{R}^d} \int_{\Theta_x} d\mu(\theta_N) \, dP_X(x) = \int_{\mathbb{R}^d} \mu(\Theta_x) \, dP_X(x).$$

By Lemma 12, we have $\mu(\Theta_x) = 0$ and therefore $\nu(S_2) = 0$ and hence $\nu(S_1) = 0$. On the other hand, Fubini's theorem again yields

$$\nu(S_1) = \int_{\Theta_{P_X}} \int_{\mathcal{N}(\theta_N)} dP_X(x) \, d\mu(\theta_N) = \int_{\Theta_{P_X}} P_X(\mathcal{N}(\theta_N)) \, d\mu(\theta_N).$$

By the definition of $\Theta_{P_X}$ we have $P_X(\mathcal{N}(\theta_N)) > 0$ for all $\theta_N \in \Theta_{P_X}$. Therefore, $\nu(S_1) = 0$ implies that $\mu(\Theta_{P_X}) = 0$. ∎

## C.3   Proofs of Section 3.5

To prove Theorem 2, we also need the following lemma.

**Lemma 13** *Let $(\theta_n)_{n \in \mathbb{N}}$ be a sequence in $\mathbb{R}^m$ converging towards $\theta_0$, i.e., $\theta_n \neq \theta_0$ as $n \to \infty$. Moreover, let $f : \mathbb{R}^m \to \mathbb{R}$ be a function and $g$ be a vector in $\mathbb{R}^m$. If*

$$\frac{|f(\theta_n) - f(\theta_0) - g \cdot (\theta_n - \theta_0)|}{\|\theta_n - \theta_0\|} \to 0$$

*holds for all sequences $(\theta_n)_{n \in \mathbb{N}}$, then $f$ is differentiable at $\theta_0$ with differential $g$.*

**Proof** [of Lemma 13] The definition of a differential tell us that $g$ is the differential of $f$ at $\theta_0$, if

$$\lim_{\delta \to 0} \frac{1}{\|\delta\|} |f(\theta_0 + \delta) - f(\theta_0) - g \cdot \delta| = 0.$$

By the sequential characterization of limits, we immediately obtain the assertion. ∎

**Proof** [of Theorem 2] Consider the following augmented networks:

$$h^{(1)}_{(\theta_D, \psi)}(X, Z) = \big( \mu^{(T)}_{\theta_D}(X), \mu^{(T)}_{\theta_D}(G_\psi(Z)) \big),$$
$$h^{(2)}_{(\theta_D, \psi)}(Z, Z') = \big( \mu^{(T)}_{\theta_D}(G_\psi(Z)), \mu^{(T)}_{\theta_D}(G_\psi(Z')) \big),$$
$$h^{(3)}_{\theta_D}(X, X') = \big( \mu^{(T)}_{\theta_D}(X), \mu^{(T)}_{\theta_D}(X') \big).$$

Without loss of generality, in the following, we only consider the network $h^{(1)}_{(\theta_D, \psi)}(X, Z)$ with inputs from $P_X \otimes P_Z$, which satisfies $\mathbb{E}_{P_X \otimes P_Z} \|(X, Z)\|^2 < \infty$.

By the expression of definition 1, we have

$$k_{RF}\big(h_{\theta_N}(X), h_{\theta_N}(G_\psi(Z)); \theta_F\big) = \frac{1}{T} \Big\langle \mu^{(T)}(h_{\theta_N}(X); \theta_F), \mu^{(T)}(h_{\theta_N}(G_\psi(Z)); \theta_F) \Big\rangle$$
$$= \frac{1}{T} \Big\langle \mu^{(T)}_{\theta_D}(X), \mu^{(T)}_{\theta_D}(G_\psi(Z)) \Big\rangle$$

where we denote $\mu^{(T)}_{\theta_D}(x) := \mu^{(T)}(h_{\theta_N}(x); \theta_F)$. Due to the linear kernel $k_L(x, y) := \langle x, y \rangle$, there holds

$$k_{RF}\big(h_{\theta_N}(X), h_{\theta_N}(G_\psi(Z)); \theta_F\big) \cdot T = k_L\big(\mu^{(T)}_{\theta_D}(X), \mu^{(T)}_{\theta_D}(G_\psi(Z))\big)$$
$$= K\big(h^{(1)}_{(\theta_D, \psi)}(X, Z)\big)$$
$$=: K\big(h_\lambda(u)\big),$$

where $u := (X, Z)$, $\lambda := (\theta_D, \psi)$, and the second equation is due to equation 16. Suppose that there exists a $\lambda_0 = (\theta_{D,0}, \psi_0)$ such that the function $(\theta_D, \psi) \mapsto (\mu^{(T)}_{\theta_D}(x), G_\psi(z))$ is differentiable at $\lambda_0$ for $P_X$-almost all $x$ and $P_Z$-almost all $z$. Then, according to Proposition 5, this statement holds for $\mu$-almost all $\theta_D \in \mathbb{R}^{|\theta_D|}$ and all $\psi \in \mathbb{R}^{|\psi|}$. Consider a sequence $(\lambda_n)_{n \in \mathbb{N}}$ that converges to $\lambda_0$, then

there exists a $\delta > 0$ such that $\|\lambda_n - \lambda_0\| < \delta$ for all $n \in \mathbb{N}$. For a fixed $u = (x, z) \in \mathbb{R}^d \times \mathbb{R}^{|z|}$, Proposition 4 states that there exists a regular function $c(u)$ with $\mathbb{E}_{\mathrm{P}_X} c(X) < \infty$ such that

$$|K(h_{\lambda_n}(u)) - K(h_{\lambda_0}(u))| \leq c(u)\|\lambda_n - \lambda_0\|$$

and consequently we have

$$|\partial_\lambda K(h_{\lambda_0}(u))| \leq c(u)$$

for $\mathrm{P}_X \otimes \mathrm{P}_Z$-almost all $u \in \mathbb{R}^{|u|}$.

For $n \in \mathbb{N}$, we define the sequence $g_n(x)$ by

$$g_n(u) = \frac{|K(h_{\lambda_n}(u)) - K(h_{\lambda_0}(u)) - \partial_\lambda K(h_{\lambda_0}(u))(\lambda_n - \lambda_0)|}{\|\lambda_n - \lambda_0\|}.$$

Obviously, the sequence $g_n(x)$ converges pointwise to 0 and is bounded by the integrable function $2c(u)$. By the dominated convergence theorem, see e.g., Theorem 2.24 in Folland (1999), we have

$$\mathbb{E}_{\mathrm{P}_X \otimes \mathrm{P}_Z} g_n(u) \to 0.$$

Moreover, for $n \in \mathbb{N}$, we define the sequence $\tilde{g}_n(x)$ by

$$\tilde{g}_n = \frac{|\mathbb{E}_{\mathrm{P}_X} K(h_{\lambda_n}(u)) - \mathbb{E}_{\mathrm{P}_X} K(h_{\lambda_0}(u)) - \mathbb{E}_{\mathrm{P}_X} \partial_\lambda K(h_{\lambda_0}(u))(\lambda_n - \lambda_0)|}{\|\lambda_n - \lambda_0\|}.$$

Clearly, the sequence $\tilde{g}_n(x)$ is upper bounded by $\mathbb{E}_{\mathrm{P}_X \otimes \mathrm{P}_Z} g_n(u)$ and therefore converges to 0. By Lemma 13, $\mathbb{E}_{\mathrm{P}_X \otimes \mathrm{P}_Z}[K(h_\lambda(u))]$ is differentiable at $\lambda_0$ with differential

$$\mathbb{E}_{\mathrm{P}_X \otimes \mathrm{P}_Z} \partial_\lambda K(h_\lambda(u)).$$

Since similar results as above hold also for the networks $h^{(2)}$ and $h^{(3)}$, and Lemma 6 in Gretton et al. (2012) states that $\mathrm{MMD}_u^2$ is unbiased, our assertion follows then from the linearity of the form of $\mathrm{MMD}_u^2$ in equation 1. ∎

## D  SAMPLES OF GENERATED PICTURES

Generated samples on the datasets CIFAR-10, CelebA, and LSUN bedroom are shown in Figure 4, 5, and 6, respectively.

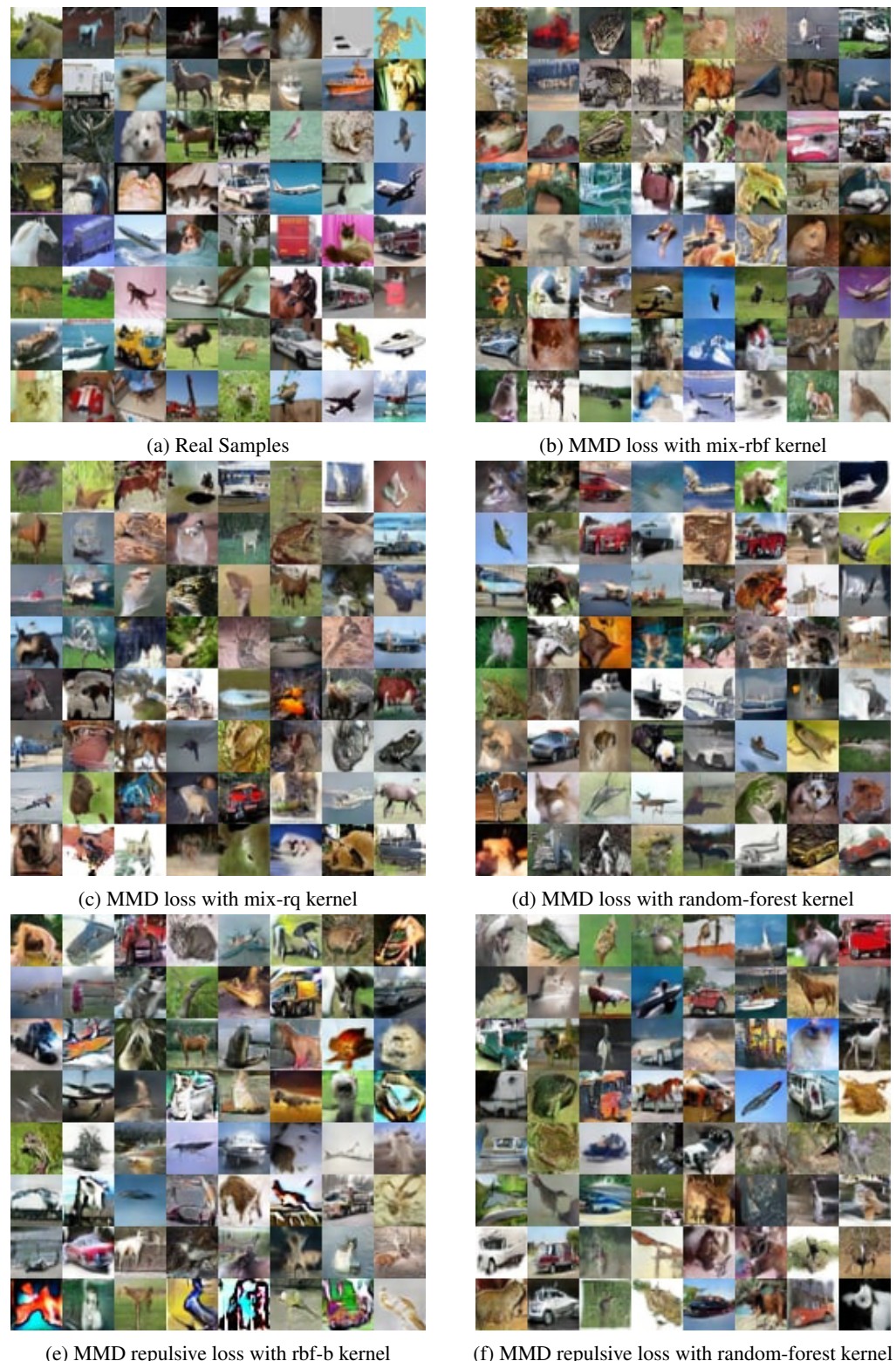

(a) Real Samples

(b) MMD loss with mix-rbf kernel

(c) MMD loss with mix-rq kernel

(d) MMD loss with random-forest kernel

(e) MMD repulsive loss with rbf-b kernel

(f) MMD repulsive loss with random-forest kernel

Figure 4: Generated Pictures with different kernels and two different loss functions on 32x32 CIFAR-10 dataset.

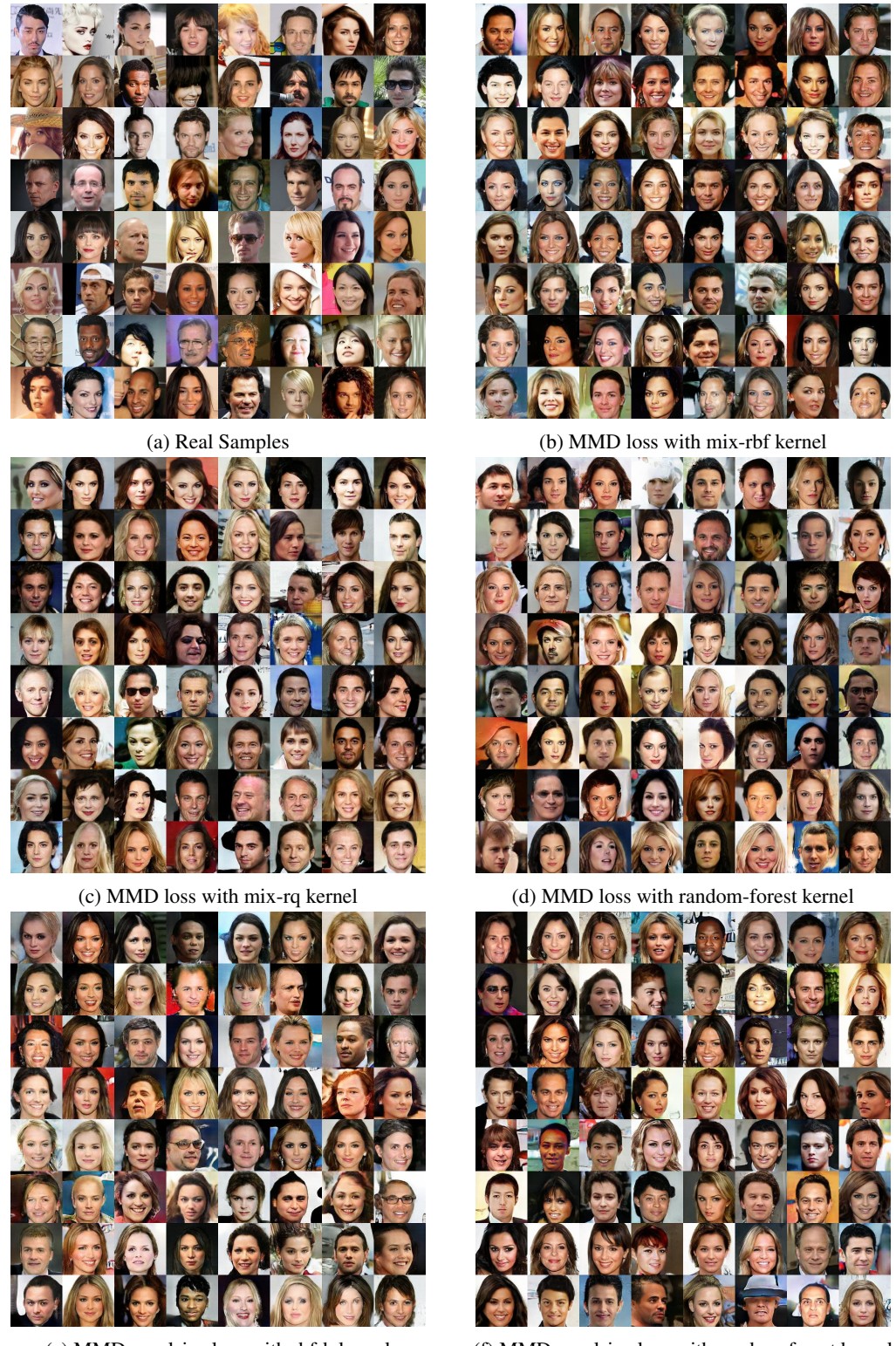

(a) Real Samples

(b) MMD loss with mix-rbf kernel

(c) MMD loss with mix-rq kernel

(d) MMD loss with random-forest kernel

(e) MMD repulsive loss with rbf-b kernel

(f) MMD repulsive loss with random-forest kernel

Figure 5: Generated Pictures with different kernels and two different loss functions on 160x160 CelebA dataset.

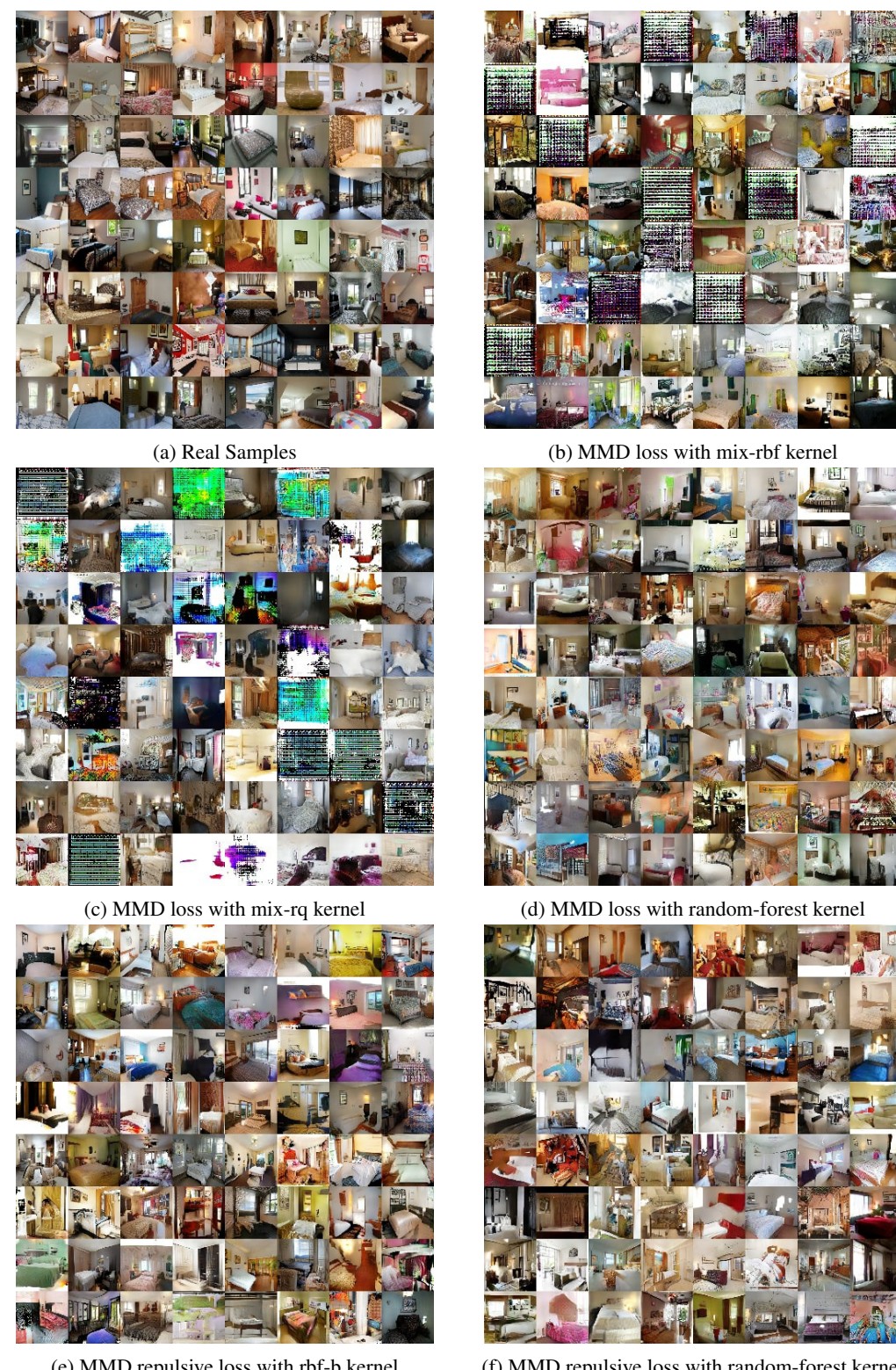

(a) Real Samples

(b) MMD loss with mix-rbf kernel

(c) MMD loss with mix-rq kernel

(d) MMD loss with random-forest kernel

(e) MMD repulsive loss with rbf-b kernel

(f) MMD repulsive loss with random-forest kernel

Figure 6: Generated Pictures with different kernels and two different loss functions on 64x64 LSUN bedrooms dataset.

