# OpenReview forum: "MMD GAN with Random-Forest Kernels"
_ICLR.cc/2020/Conference — Reject_

### Official Review · AnonReviewer3 · 2019-10-22
**Official Blind Review #3**

**Rating:** 3

**Review:**

Overview:
The paper propose an MMD GAN extension via using Random forest Kernel.  Instead of using Gaussian kernel on the top of the learned embeddings from the discriminator, it combines existing deep forests kernels. The theory of being differentiable is carefully studied (to prove zero measure) and the experiments are well conducted.

1.  Some important  references are missing.  One very related paper is

* Li et al., Implicit Kernel Learning,  AISTATS 2019.

That paper is using the same idea to learn to manipulate the random features on the top of the learned embedding.  The main difference between it and the proposed algorithm is they use MLP parameterization instead of the tree-based model.    Also, the deep forest model can be treated as a sparse neural network, does it have more advantage over Li et al., (2019)? given they use simple dense MLP.   Please at least discuss the similarity and difference in the rebuttal and update the draft correspondingly.  I would even encourage the author to empirically compare with it in the camera ready version.  It would be interesting to see which parameterization is better in this space.

There are also other recent MMD GAN extensions should be cited in the discussion, such as
* On gradient regularizers for MMD GANs.

2. For the theory part, based on Binkowski (2018), the gradients for the generator parameters should be biased. Could you discuss it with Theorem 2?

3. For most MMD GAN results, one important property in Li et al., (2017),  Arbel et al., (2018) and Li et al., (2019) is weak* topology.  Does the proposed Random Forest MMD GAN also has that property? In Li et al., (2019), they need some condition to ensure that, how's case in the proposed algorithm?

**Experience Assessment:**

I have published in this field for several years.

**Review Assessment: Checking Correctness Of Derivations And Theory:**

I assessed the sensibility of the derivations and theory.

**Review Assessment: Checking Correctness Of Experiments:**

I carefully checked the experiments.

**Review Assessment: Thoroughness In Paper Reading:**

I read the paper at least twice and used my best judgement in assessing the paper.

---

### Official Review · AnonReviewer1 · 2019-10-22
**Official Blind Review #1**

**Rating:** 1

**Review:**

Theorem 2 and its proof are plagiarised: they are rephrased and reorganized formulation and proof of Theorem 1 of [1], while being presented as authors' own work.

Although the assumptions are slightly different (random forest kernels vs general kernels), core of the proof is the same, including notation and its split into Lemmas and helper Theorems. In particular:
- formulation is the same (even use of MMD_u is copied, while not being defined before),
- main proof of Theorem 2 (p.25-26) is the proof of Corollary 3 of [1] followed by the proof of Theorem 5 of [1],
- Lemma 13 is Lemma 3 of [1],
- Definition 3 in Appendix B.2 is the same as Assumption D of [1] (Appendix C.2),
- Proposition 4 and it's proof (p. 21) are the same as Lemma 2 of [1].

[1] Mikołaj Bińkowski, Dougal J. Sutherland, Michael Arbel, and Arthur Gretton. Demystifying MMD GANs. International Conference on Learning Representations, 2018.

**Experience Assessment:**

I have published one or two papers in this area.

**Review Assessment: Checking Correctness Of Derivations And Theory:**

I carefully checked the derivations and theory.

**Review Assessment: Checking Correctness Of Experiments:**

I carefully checked the experiments.

**Review Assessment: Thoroughness In Paper Reading:**

I read the paper thoroughly.

---

### Official Review · AnonReviewer2 · 2019-10-22
**Official Blind Review #2**

**Rating:** 1

**Review:**

This paper proposed to utilize the random-forest kernel into MMD GAN.  The experiments are conducted on CIFAR-10, CelebA and LSUN datasets.

The method is not novel. Both MMD GAN and the random-forest kernel have been well explored. Combining them together is considered as an extension. For the theory, the paper only provides the unbiasedness analysis. It is not clear to me whether this kernel is better than other MMD GAN variations. It is not clear how the claimed flexibility comes from.

Regarding the experiments, it only compares with very basic baselines and the results are not significantly better. It would be better to include stronger baselines (Wang et al., 2019, Binkowski et al., 2018).

The writing of the paper is poor. with several typos. Moreover, as mentioned by reviewer #1,  theorem 2 and its proof are plagiarised.

Overall, I think the paper is a clear reject.

**Experience Assessment:**

I have published in this field for several years.

**Review Assessment: Checking Correctness Of Derivations And Theory:**

I assessed the sensibility of the derivations and theory.

**Review Assessment: Checking Correctness Of Experiments:**

I carefully checked the experiments.

**Review Assessment: Thoroughness In Paper Reading:**

I read the paper thoroughly.

---

### Decision · Program_Chairs · 2019-12-19

**Decision:**

Reject

**Comment:**

Reviewers raise the serious issue that the proof of Theorem 2 is plagiarized from Theorem 1 of "Demystifying MMD GANs" (https://arxiv.org/abs/1801.01401). With no response from the authors, this is a clear reject.